# Resilience of the Eastern African electricity sector to climate driven changes in hydropower generation

Vignesh Sridharan [1], Oliver Broad [1,2], Abhishek Shivakumar[1], Mark Howells[1], Brent Boehlert [3,4], David G. Groves[5], H-Holger Rogner[1,6], Constantinos Taliotis [7], James E. Neumann[3], Kenneth M. Strzepek[3,4], Robert Lempert[5], Brian Joyce[8], Annette Huber-Lee[8] & Raffaello Cervigni[9]

Notwithstanding current heavy dependence on gas-fired electricity generation in the Eastern African Power Pool (EAPP), hydropower is expected to play an essential role in improving electricity access in the region. Expansion planning of electricity infrastructure is critical to support investment and maintaining balanced consumer electricity prices. Variations in water availability due to a changing climate could leave hydro infrastructure stranded or result in underutilization of available resources. In this study, we develop a framework consisting of long-term models for electricity supply and water systems management, to assess the vulnerability of potential expansion plans to the effects of climate change. We find that the most resilient EAPP rollout strategy corresponds to a plan optimised for a slightly wetter climate compared to historical trends. This study demonstrates that failing to climate-proof infrastructure investments can result in significant electricity price fluctuations in selected countries (Uganda & Tanzania) while others, such as Egypt, are less vulnerable.

---

[1] KTH – Royal Institute of Technology, Unit of Energy Systems Analysis, Office K514, Brinellvägen 68, 100 44 Stockholm, Sweden. [2] UCL Institute for Sustainable Resources, University College London, Central House, 14 Upper Woburn Place, London WC1H 0NN, UK. [3] Industrial Economics Inc., 2067 Massachusetts Ave, Cambridge, MA 02140, USA. [4] Massachusetts Institute of Technology, 77 Massachusetts Avenue, Cambridge, MA 02139, USA. [5] RAND Corporation, P.O. Box 2138, 1776 Main Street, Santa Monica, CA 90407-2138, USA. [6] International Institute for Applied Systems Analysis, Schlossplatz 1, A-2361 Laxenburg, Austria. [7] The Cyprus Institute, 20 Konstantinou Kavafi Street, 2121 Aglantzia, Nicosia, Cyprus. [8] Stockholm Environment Institute-US Centre, 11 Curtis Avenue, Somerville, MA 02144-1224, USA. [9] World Bank, 1818 H Street, NW Washington, DC 20433, USA. Correspondence and requests for materials should be addressed to V.S. (email: vsri@kth.se)

The uncertainty in projections of global precipitation and temperature has been widely discussed in the literature[1,2]. For the period 2010–2050, the average Climate Moisture Index (CMI), a measure of aridity that combines the effect of rainfall and temperature—across the different General Circulation Model (GCM) outputs from the two Coupled Model Inter-Comparison Projects (CMIP3 and CMIP5)—reflects a wide range of uncertainty for the African continent's major River Basins[3]. This potential range of future climates has long-term impacts on the electricity generation sector; with water levels for hydropower generation and cooling requirements in thermoelectric power plants being the most affected[4]. The term climate, in this assessment, refers to a consistent set of temperature and precipitation projections. Amongst the CMI values for the illustrated African basins, the sub-basins of the Nile River display high levels of uncertainty (Nile Equatorial Lakes) and some of the driest projections (Eastern Nile)[5]. Despite the Eastern Nile sub-basin contributing almost 80% of the river's total annual flow, the White Nile (originating from the Nile Equatorial lakes) provides nearly all of the flow during approximately four months in a year—which make its contribution significant. Countries like Uganda depend 100% on the White Nile for its hydropower infrastructure.

A majority of countries in the Nile basin region have low electrification rates[6]. With an expected increase of 50% in the population by 2030[7] and accounting for the well-documented link between electrification and development, the countries have an imperative to improve electricity access and reliability. With a mandate to increase access to electricity, and develop a strategy for the optimum use of energy and financial resources in the region by stimulating cross-border collaboration, an Inter-Governmental Memorandum of Understanding (IGMOU) was signed by ten countries that lie along the Nile River, to form the EAPP.

The EAPP[8] consists of Rwanda, Djibouti, Tanzania, Kenya, Burundi, Uganda, Sudan, Ethiopia, Egypt, and Libya. South Sudan was part of the EAPP when united with Sudan and is expected to re-join the group soon; hence, Sudan—in this article —refers to both present-day Sudan and South Sudan. All except Libya lie either partially or entirely inside the Nile River Basin, hence Libya is not considered in this study. In addition to the growing residential water demand and irrigation water requirements, a significant share of the electricity generation infrastructure in the region relies on water from the River Nile.

With an installed hydropower generation capacity of 8.7 GW and additional planned capacity of 22 GW in the pipeline, hydro infrastructure contributes to more than 50% of the electricity generation in many EAPP countries and is expected to continue playing a decisive role in the Power Pool's energy system[9,10]. Most of these investments in large hydropower plants are made under the assumption that precipitation patterns will resemble historical trends. Therein lies a severe risk. Any change in water availability could leave this infrastructure stranded or result in lost opportunity cost in not taking advantage of higher water availabilities.

In recent literature, expansion of electricity infrastructure in the EAPP countries has been discussed without giving climate uncertainty due consideration. The latest EAPP master plan[10] analyses the sensitivities in hydropower production for specific dry, wet and driest year combinations based on historical hydro inflows. Kammen et al.[11] discuss the inherent risks involved in relying heavily on hydropower expansion to improve energy access in East Africa. Though detailed, these studies consider the energy system in isolation from other water reliant systems. They do not take into consideration other sectors, which compete for the same water resource, namely: agriculture, industries and domestic households. The formulation of policies and infrastructure investment decisions without considering all relevant sectors, coherently and consistently, could have severe impacts[12]. In the case of Mauritius, Howells et al.[13] found that such oversights could lead to inconsistent policy advice with counter-intuitive repercussions. A similar study conducted on a regional scale for the Iberian Peninsula[14] illustrates the need for such inter-sectoral studies.

Aggressive urbanisation, irrigation expansion plans and variable release from upstream hydropower plants across the power pool could result in fluctuating water availability for downstream countries. This may result in geopolitical conflicts; which are not new to the Nile River Basin[15–17]. Therefore, there is a need to examine energy and water use coherently to assess power pool infrastructure resilience to the effects of a changing climate. To address this issue, we develop a framework, soft-linking two models —first, a long-term electricity expansion model of the EAPP, formulated in an open-source platform[18,19]—second, a water systems management model of the Nile River Basin developed using a widely used water balance management framework[20,21]. We then calibrate the water system model to historical stream flows using data from multiple locations along the Nile River. The calibrated model is then subjected to harmonised precipitation and temperature projections using downscaled, bias-corrected, GCM outputs from the CMIP3 and CMIP5 suite of models[22]. Two decision criteria—a mini-max regret criterion and a variant of the domain criterion—are employed to identify the most resilient adaptation strategy for the EAPP.

Our results indicate that the most resilient EAPP rollout strategy corresponds to a plan optimised for a slightly wetter climate compared to historical trends. We highlight the importance of cross-border electricity trade in mitigating climate change impacts on the power system of the EAPP. Failing to climate-proof electricity infrastructure in the EAPP countries could result in fluctuations in consumer electricity prices, which might be a hindrance to economic development in the region.

## Results

**Baseline and perfect foresight expansion strategies.** Figure 1 illustrates the geographic scope of this study. The expansion plans from the Program for Infrastructure Development in Africa (PIDA)[23] along with country-level energy, water, and irrigation master plans were combined to form a database, which served as reference data for the modelling framework; referred to as PIDA+. We developed a sequence of runs—of the energy and water models—for a range of climates to assess expansion plan performance. This sequence consisted of three types of model runs; first, a baseline run that simulates a case where the climate follows a historical pattern. Second, perfect foresight adaptation (PF) runs that identify the cost-optimal expansion plan for each climate. The PF runs simulate a case where the energy planners know which future climate will occur and have the ability to adapt: (a) either up- or down-size selected hydropower infrastructure while (b) investing in alternative technologies to ensure the security of supply and reduce the cost of electricity generation, and (c) exploring electricity trade options with neighbouring countries. Third, no-adaptation (NA) runs that fix the infrastructure rollout from each PF run and simulate the impact of different climates.

Our baseline results indicate that, if the future climate is expected to be similar to historical patterns, the investments in hydropower for the PIDA+ expansion plan (2015–2050) are expected to be in the range of USD$_{2010}$ 75 billion. Comparatively, the entire power sector investments for the same period are expected to reach USD$_{2010}$ 2.3 trillion. These numbers are

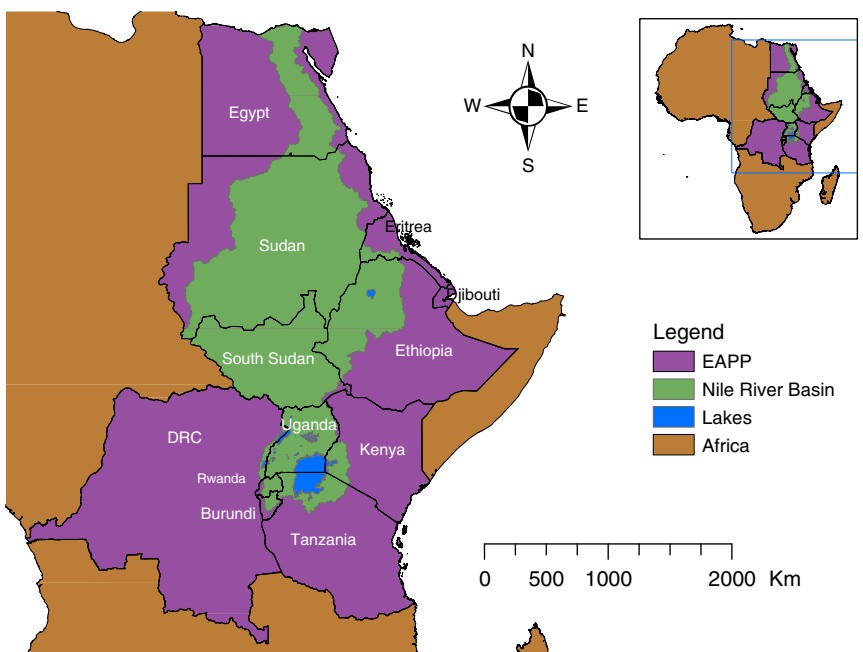

**Fig. 1** The Nile River Basin and the EAPP countries. The Nile River Basin spreads across countries in East Africa, which constitute an institution called the Eastern African Power Pool (EAPP)

interesting considering that, in the same period, the share of hydropower in the total electricity generation mix is significant, close to 21%. Despite the share of natural gas in total electricity generation dropping from 72% in 2015 to 52% in 2050, it is still expected to play an essential role in the power pool's electricity mix. Egypt is expected to contribute up to 76% of total natural gas consumption in 2050, as it has access to potentially vast local natural gas resources. It is assumed that this gas can be extracted at competitive costs. Towards the second half of the modelling period (post-2030), higher electrification rates would also see countries like Ethiopia, Tanzania, and Uganda taking advantage of local renewable-based electricity generation potential, including hydropower.

Figure 2 highlights the importance of both non-hydro renewables and nuclear power in the EAPP's generation mix. Note that the latter is restricted to Egypt where a high domestic baseload electricity demand can accommodate the bulk generation of current nuclear power plants. Finally, it is salient that the cost of electricity generation increases, both nationally and on a power pool scale, due to higher shares of fossil fuels in systems that were previously hydro-dependent, e.g., Tanzania. This highlights the potential for beneficial effects from increased renewables in resource-rich countries, particularly for hydropower, provided water availability remains reliable. Results for hydropower dependent countries, in Fig. 2, reflect the impact of variation in annual water availability on the cost of electricity generation. The total installed capacity and the electricity generation mix for all analysed climates, in each of the EAPP countries, highlight the importance of hydropower in this region of Africa (Supplementary Figures 1–14).

However, infrastructure development faces deep uncertainty, as the future climate remains unknown. Our NA model runs address this uncertainty by maintaining the PIDA+ roll-out constant while varying future levels of water availability under different climates. Only a limited set of expensive diesel-based electricity generation options is allowed to contribute to the loss of hydro generation. Large build-out of new coal or gas is not permitted. In our labelling, we refer to dry or wet climate futures. Note that these are cumulatively drier or wetter in the period 2015–2050 for the entire Basin, rather than systematically dry or wet across all months and years (Supplementary Figures 15–16).

**Climate resilience of expansion strategies.** The non-adaptation of the electricity sector is expected to affect different EAPP countries to varying degrees. Countries like Egypt and Kenya, which are expected to rely on gas, geothermal, and coal for the security of their electricity supply, are barely affected. In the NA scenario for the driest climate, Egyptian consumers are expected to face a cumulative change in the expenditure of approximately +1% over the modelling period (2017–2050) when compared to the baseline. This is due to the low share of hydro-generation in Egypt. Conversely, Uganda and Tanzania could expect costs to vary across ranges of −18% to +28%, and −5% to +23% respectively between the wettest and driest climates when compared to the baseline (cumulatively between 2017–2050). Figure 3 compares the cost of electricity generation across three strategies: the baseline expansion strategy under historical climate trends, no–adaptation strategy under the driest climate and the most resilient adaptation strategy. The most resilient strategy has the lowest value among the calculated maximum financial regrets for all the analysed climate futures. It illustrates the yearly variation in the cost of electricity generation for Uganda, Tanzania, and Egypt. Also noticeable are the substantial fluctuations in electricity cost under the no adaptation scenarios. By emphasising the variability of climatic impacts in different countries, it inherently reflects the resilience or vulnerability of the current electricity infrastructure to climate change (Supplementary Figures 17–19).

An important insight from the analysis is that the most resilient strategy is the one planned for a wetter climate. This would not have been intuitively expected prior to this analysis; the strategy has the lowest value for the maximum regret across the different climate futures (Supplementary Note 1). Note

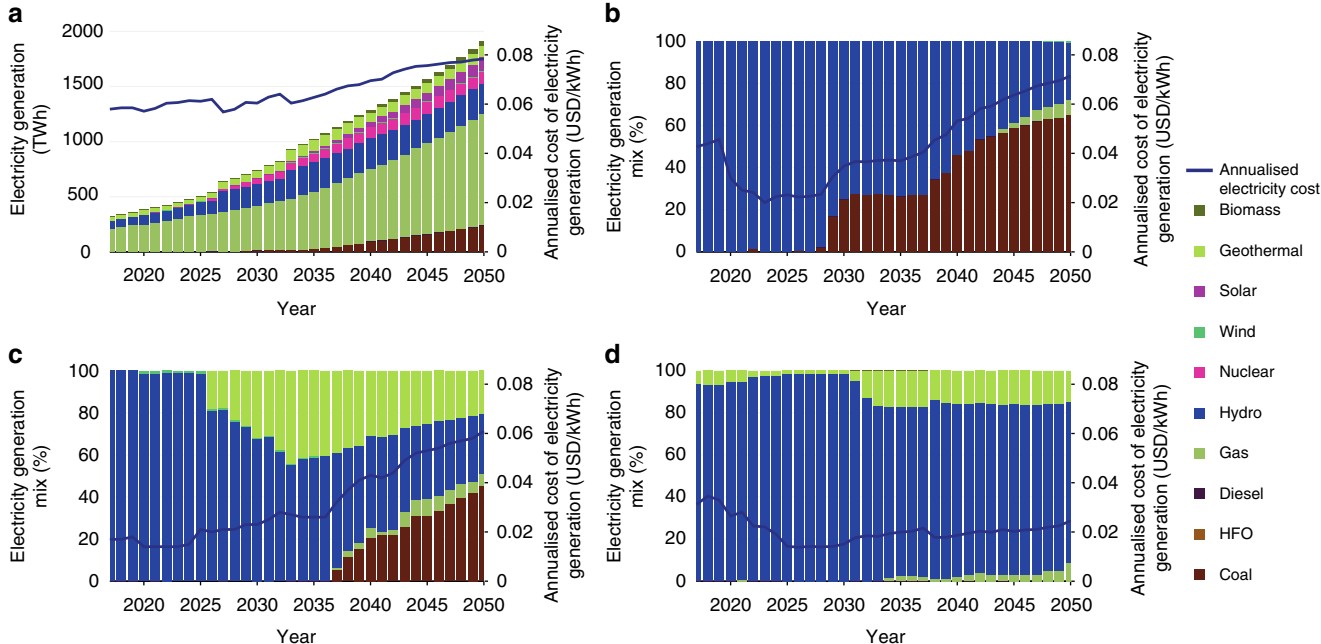

**Fig. 2** Baseline electricity generation in the EAPP. **a** Combined electricity generation in the EAPP (TWh) plotted along with the cost of generating electricity (USD/kWh). **b** Electricity generation mix in Tanzania (%) and the cost of electricity generation (USD/kWh). **c** Electricity generation mix in Ethiopia (%) and the cost of electricity generation (USD/kWh). **d** Electricity generation mix in Uganda (%) and the cost of electricity generation (USD/kWh). The electricity generation graphs correspond to the baseline scenario, where the precipitation and temperature projections correspond to a historical climate pattern in the respective countries

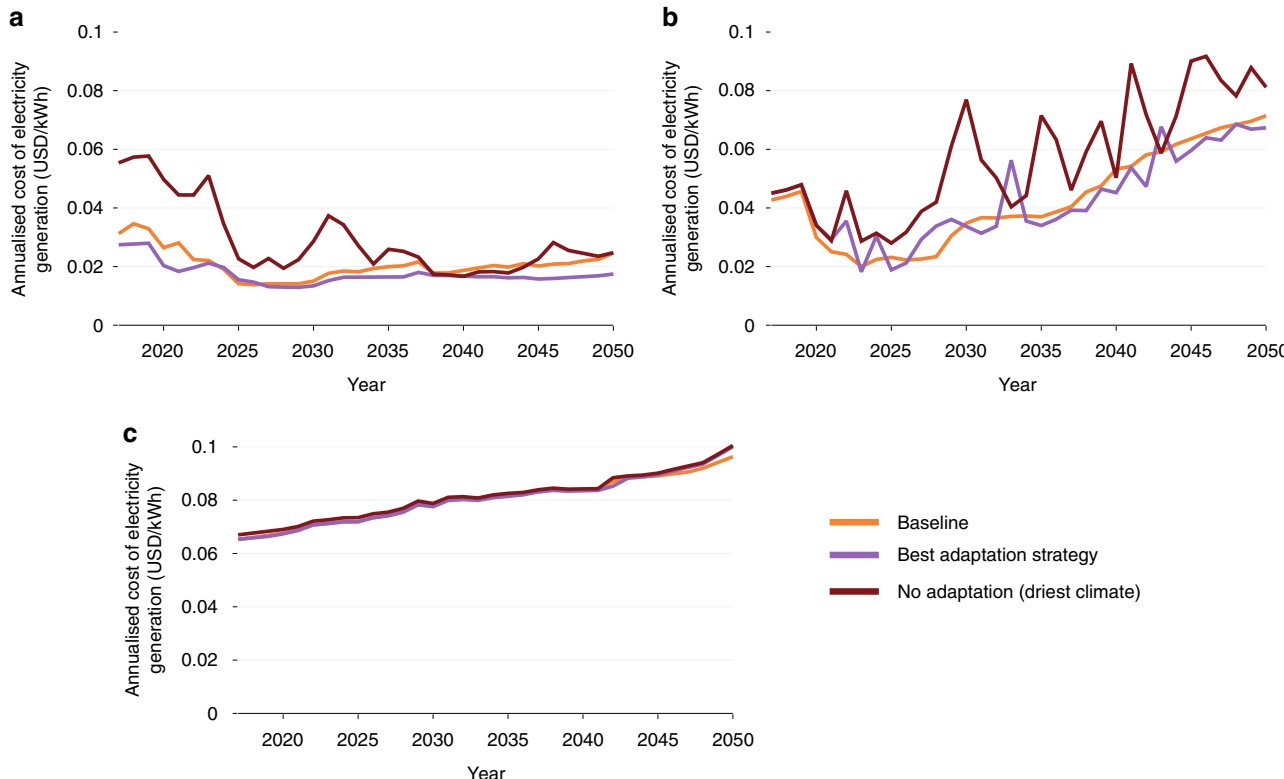

**Fig. 3** Impact of climate on the cost of electricity generation. The figures illustrate the annualised cost of electricity generation (USD/kWh) for three scenarios: the baseline, a no-adaptation strategy for the driest climate and the most resilient adaptation strategy for three countries—**a** Uganda, **b** Tanzania, **c** Egypt

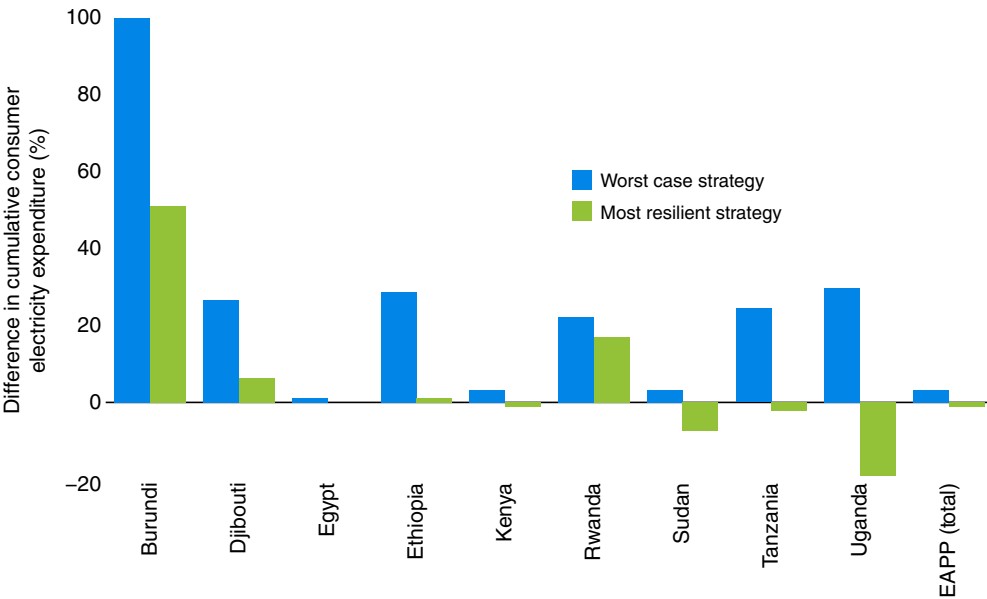

**Fig. 4** Impact of adaptation strategies on electricity expenditure. The cumulative consumer electricity expenditure under the most resilient and the worst (no adaptation-driest climate) strategy are compared against the baseline. The zero line (0%) refers to the expenditure in the baseline. The accumulation period is from 2017–2050

that the most resilient strategy includes adaptation across all three sectors—energy, water, and agriculture. In this paper, we highlight only the performance indicators relevant to the electricity infrastructure. Full results from the study including other performance indicators are available in Cervigini et al.[3].

Figure 4 compares the cumulative consumer electricity expenditure for the most resilient strategy to the worst-case scenario (NA strategy for the driest climate) expressed as a percentage difference from the baseline for selected EAPP countries. Average change on a power-pool scale is small and reflects the fact that Egypt—the country with the highest electricity demand in the EAPP—tends to drive power-pool level performance indicators. Ethiopia and Uganda stand to benefit the most from choosing the most resilient strategy over their respective worst case options, with cumulative electricity expenditure, over 2017–2050, dropping from +27% to +1% and from +28% to −18.5% respectively relative to the baseline. These translate to savings of approximately USD$_{2010}$ 19 billion and USD$_{2010}$ 4 billion in consumer expenditure in Ethiopia and Uganda respectively. In the case of Tanzania, despite having sizable amounts of fossil fuel in the generation mix, a drier climate leads to increased electricity imports as large quantities of fossil investment in the short period is not allowed. These risk causing electricity price fluctuations if proper import power purchase agreements and sufficient transmission infrastructure are not in place.

**Cross-border electricity trade for climate change mitigation**. Results across all scenarios place Ethiopia as a major electricity exporter by 2040, trading on average 30 TWh annually or 15% of its yearly electricity generation. Figure 5 illustrates the cumulative, power pool level, electricity trade (2017–2050) under PF driest and wettest climates, highlighting the significant impact that variations in regional climate have on net power flows. Under the driest climate (Fig. 5a), Ethiopia would export about 415 TWh to Sudan which, in turn, exports 245 TWh (59%) to Egypt. Under the wettest climate (Fig. 5b), though Ethiopia exports about 600

TWh of electricity to Sudan only 108 TWh (18%) are further directed to Egypt. This represents a drop both in absolute flow to Egypt, now relying on 228 TWh additional domestic hydropower, and in relative terms, increasing Sudan's levels of net imports. It also shows that Sudan will absorb available and cheap power to displace its gas-based electricity generation and harmonise its domestic electricity cost. These results support the message that irrespective of climate, the availability of electricity trading infrastructure stands to play a significant role in balancing electricity prices within the EAPP.

**Discussion**

The effects of a drier climate in upstream Nile countries like Uganda, Tanzania and Burundi could be mitigated when the Democratic Republic of Congo's Grand Inga complex is developed, and investments are made in inter-country transmission infrastructure[24]. The climate-related risk involved in establishing this infrastructure is expected to be low as a majority of GCM projections suggest the Congo basin to be less affected by potential drying. This is evident from the comparison of the CMI values for the Congo Basin to the other analysed River Basins (Supplementary Figure 20).

The results from this study should, however, be treated with caution. Despite implementing a detailed representation of both electricity and water infrastructure, some uncertainties such as future electricity demand and fuel prices are not explored. Spalding-Fecher et al.[25,26] take the case of the Southern African Power Pool (SAPP) to illustrate that applying an ambitious sustainable pathway similar to SSP1 (Shared Socio-Economic Pathways) would result in 8 out of the 12 Southern African nations shifting from coal and hydro-based generation to solar PV. A slightly less optimistic outlook could be expected for the EAPP after taking into consideration grid integration and storage costs. Also, fluctuations in fossil fuel prices have a significant impact in the region, especially with Egypt's dependence on natural gas. With prospects of new gas finds (Zohr gas field)[27], EAPP dynamics could be altered to rely on Egyptian gas as security

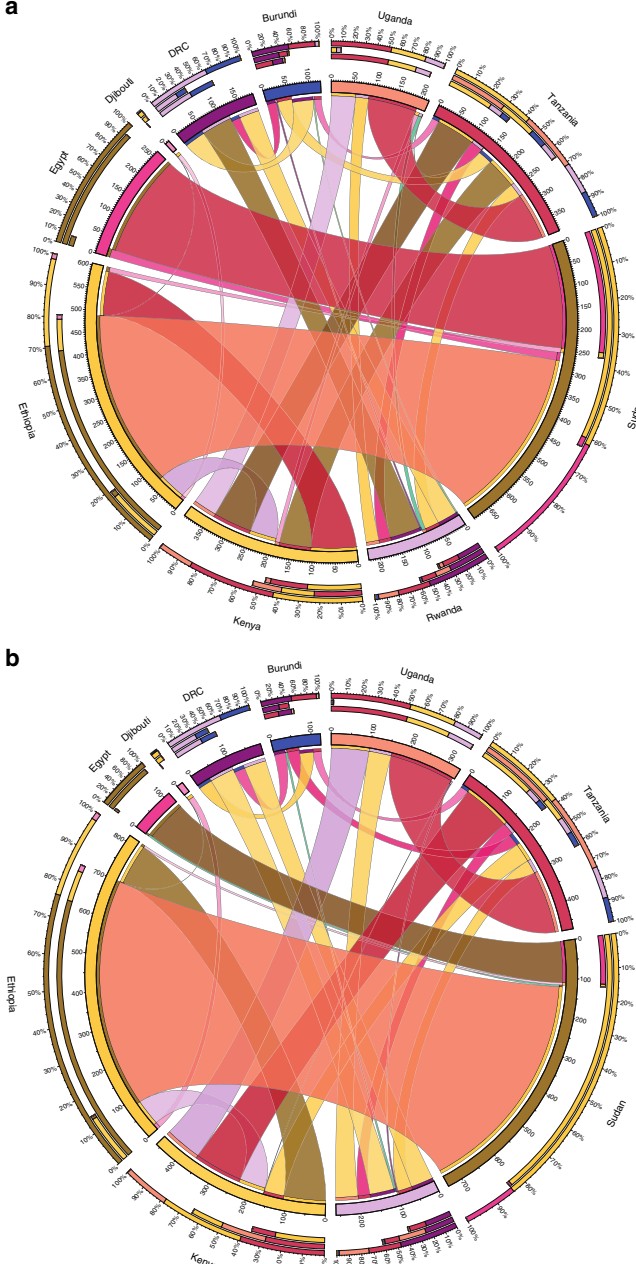

**Fig. 5** Electricity trade under perfect foresight strategies. **a** Cumulative (2017–2050) electricity trade under perfect foresight driest climate. **b** Cumulative (2017–2050) electricity trade under perfect foresight wettest climate. The outer circle highlights the different countries of the EAPP. There are three bars (percentage mix) corresponding to the outermost circle. In the order of outermost to the innermost bar, the first refers to the exported electricity; the second refers to the imported electricity and the third to the difference. The colour of the bar (arc), on the inner circle, is specific to the country where electricity is exported or imported from. The scale on the circumference of the inner circle shows the absolute values of electricity trade in TWh. Each flow band denotes the cumulatively traded electricity from country A to country B. Flow bands attached to a country's inner circle represent exports from that country and vice-versa

against a drop in hydropower generation in the event of climate change. The price of other fossil fuels is, however, not stationary either. Though we consider an increasing fossil fuel price derived

from the EAPP master plan, a sudden change in oil and coal prices in countries like Tanzania and Kenya could, despite the cost assumptions in the model, change the generation mix significantly towards renewable-based power. In this study, for all generic power plants, we consider a capital cost that does not change over time in the modelling system. The values correspond to the initial technology-specific capital costs used in all four scenarios mentioned in IRENA's analyses of the SAPP[28]. Hence, as expected, implementation of learning rates for renewable technologies will affect the generation mix of the power pool, as illustrated in the supplementary material (Supplementary Note 2).

An analysis reveals that introducing a reduction in technology costs due to learning rates will replace electricity generation from fossil fuel power plants and not from hydropower. The analysis demonstrates that electricity trade based on inexpensive hydropower infrastructure will continue to play a significant role in the power pools generation mix (Supplementary Figures 21–24). Of course, the effects of a progressively dry climate could result in higher—non-hydro—renewable penetration, which needs to be explored in detail. Despite the relevance of uncertainties discussed above, it was a deliberate choice of this study to keep capital costs constant to help with tractability of climate change impacts on the electricity infrastructure, with a primary focus on hydropower plants.

Though the study considered an advanced set of renewable technology options along with site-specific costs for hydropower plants, it did not explore new flexible options, such as grid-level storage. This could, however, be game changing in mitigating climate change by absorbing electricity from intermittent renewables on a large scale and dispatching it when necessary[29]. Another fundamental issue that is beyond the scope of this study is the geopolitical debate in the East African region regarding the sharing of waters of the River Nile. Egypt had temporarily dropped out of the power pool due to regional conflicts related to water sharing[30,31]. We would, also, like to highlight that this study does not take the variability in weather-induced solar and wind power generation into consideration to maintain the tractability of climatic impact on hydropower generation. Though not explored in this article, in addition to the electricity infrastructure's resilience, water stress that could potentially arise from any of the analysed climate futures could also be calculated with this framework. Those stresses could then be evaluated (using different allocation priorities) to determine the vulnerability of the users. Moreover, with defined user vulnerability, the efficacy of specific resilience measures could be evaluated.

To conclude, it is interesting to note that both decision criteria used in this study converge to the same resilient strategy (Supplementary Note 1). This study suggests that, for the analysed climate futures, the regret for energy planners of the EAPP resulting from planning for a drier climate future is higher than the one resulting from increasing investment in hydropower; that is, planning for a slightly wetter than realised climate. Of course, the results will need validation on a local scale from country-level analysis with higher spatial and temporal resolution. Nevertheless, from the climate futures analysed in this study, the need to climate-proof electricity infrastructure in the EAPP is clear. The region requires a rapidly expanding electricity supply infrastructure to enable access to electricity services for all and support the EAPP countries in their path towards industrialisation. Failure to consider alternative climate futures may result in vulnerable electricity expansion planning, leading to fluctuations in consumer electricity prices.

## Methods

**A framework to link energy and water models**. This study is based on a framework involving two analytical modelling tools. The first is a detailed energy systems model of the Eastern Africa Power Pool (EAPP) covering the electricity generation mix in each of the constituent countries. The second is a water systems model of the Nile River basin. The two were soft linked to assess the resilience of water and energy infrastructure to a changing climate. WEAP, the Water Evaluation, And Planning system is used to develop the detailed water systems model (henceforth referred to as the water model) of the Nile River Basin[20] and is a widely applied integrated water management model refined over the past 20 years. The electricity sector expansion model (henceforth referred to as the energy model) was developed using OSeMOSYS, the Open Source energy MOdelling SYStem[18] (Supplementary Note 3). It considers a comprehensive technology portfolio inclusive of region-specific renewable and fossil fuel based electricity generation options (Supplementary Figure 25). Existing, committed and planned electricity generation technologies with site-specific representation for hydropower plants are included in the power pool model (Supplementary Figures 26–29, Supplementary Note 4). New hydroelectric plants are always chosen from the list of identified projects; no generic hydropower plant options are considered. Other generation options are represented using an aggregate total for their category (e.g., onshore wind, grid-connected solar photovoltaics (PV), rooftop solar PV to name a few) and were afforded varying flexibility to install new capacity under different scenarios. The OSeMOSYS framework represents all generation and transmission infrastructure using separate technologies. The EAPP energy model includes over 150 site-specific hydropower plants as well as additional electricity generation options (Supplementary Table 1). In total, this represents approximately 500 technologies forming a ten-country power pool model. The energy model has a detailed temporal resolution of 48-time slices (steps) and captures monthly water availability, at the hydropower plant level, across the different precipitation patterns of the river basin[32]. Renewable energy potentials for each country are obtained from the latest estimates by the International Renewable Energy Agency (IRENA)[33]. Each of the countries can trade electricity-using technologies that represent the transmission infrastructure. Any expansion of the existing transmission system is chosen from a list of identified or planned projects. Hydropower plants represented in the energy model are also included in the water model with or without reservoirs, as applicable. In addition, the water model contains a detailed representation of the other water and agriculture (irrigation) infrastructure. Flow regulation constraints from country-specific water management reports are also taken into account.

**Climatic representation**. To simulate different climates, the water model is parameterised using bias-corrected and spatially disaggregated (BCSD) precipitation and temperature projections from an ensemble of climate projections. GCM outputs exhibit systematic errors (biases) due to the limited spatial resolution, simplified physics and thermodynamic processes to name a few. Bias-correction is one of the commonly used calibration techniques to minimise the bias in climate model outputs, which are inputs for models that assess the impact of climate change[34]. The projections use the results from two classes of climate models: namely the CMIP3 and CMIP5 from the IPCC Assessment Report 4 (AR4) and 5 (AR5) respectively[35,36]. The baseline uses data from the Terrestrial Hydrology research group at Princeton University for the period from 1948–2008[37,38]. The BCSD process results in 121 different climate futures spanning three emission scenarios from the AR4 namely: A1, A1B, A2; and two representative concentration pathways from AR5 namely: RCP4.5 and RCP8.5. The two RCPs are chosen to represent medium and high emission scenarios.

Running the water and energy models over 121 different futures, with monthly time resolution for the former but longer time intervals for the latter is computationally intensive. Hence, a representative subset of six climate futures is therefore chosen to represent a good sample of the range of precipitation and temperature outcomes inferred from the 121 climates (Supplementary Note 5). The climate futures are ranked based on their CMI averaged over the modelling period, and a simple algorithm is developed to choose six cases which span over different percentile ranges (Supplementary Table 2)[22,39].

**Evaluation of climate resilience**. The models are linked using the water availability for hydropower generation provided by the water model on a plant-specific basis; this availability is translated into capacity factors for the electricity expansion model under each climate. In turn, a separate Matlab based optimisation framework[40] uses the cost of generating electricity from the energy model to produce a list of adaptation options for up- or down-sizing selected hydropower infrastructure. In this framework, water requirements for domestic use and agriculture take precedence over hydropower generation. As a result, adjusted power plant capacities in the water model cascade into updated values for hydropower capacity factors in the energy model (Supplementary Note 6).

The framework for evaluating the impacts of climate on the energy sector involves three types of energy model optimisation runs, detailed as the following —first, the baseline run, where the water and energy models are run assuming that historical climate patterns repeat over the modelling period. All planned infrastructure investments that are either committed or under construction (according to PIDA+) are assumed available for electricity generation in their respective years. To compensate for shortfalls in the electricity supply, the model is free to choose from a limited list of country-specific electricity generation options. Moreover, countries are also able to import electricity from their neighbours through existing or planned transmission lines. No unplanned transmission lines are made available to provide conservative estimates of trade capacities. Second (in perfect foresight (PF) adaptation runs), the energy model is run for the six chosen climate scenarios. The runs have perfect foresight, and the energy system can adapt to changes in water availability in the respective climates. The water availability is communicated between the water and the energy model using capacity factors. Each of these runs has different monthly and plant-specific capacity factors for each year during the modelling period. Hence, in dry years, the models can choose electricity generation options that minimise the overall system cost based on resource availability in the respective countries. That is, the electricity expansion model can substitute reduced hydropower availability with other electricity generation options and trade in a dry climate future and vice-versa. This soft-linking routine is iterated twice for each climate to converge towards an optimal hydropower infrastructure within the realm of possible size configurations. Third (in no-adaptation (NA) Runs), the energy model is run to simulate the effects of not adapting. They consider the infrastructure roll-out resulting from a PF run (for each climate scenario) and assess its performance across the five chosen climate futures it is not designed for. The PF expansion plan for each climate future is thus an NA strategy for the remaining five climates. In each alternate future, shortfalls in generation due to lack of water for hydropower can be satisfied only with rapidly deployable, yet expensive, diesel-based generation options. These runs, therefore, are used to evaluate the performance of each PF infrastructure rollout across the ensemble of climates.

For each climate, the Net Present Value (NPV) of cumulative consumer expenditure from each of the NA runs is compared against the best expansion strategy for that climate (PF strategy) to estimate the so-called financial regret of ill-fitting infrastructure portfolios (Supplementary Tables 3 and 4). It must be noted that these regret numbers are calculated across all climate futures using information from both the power sector expansion model (EAPP) and the water management (WEAP) model. Summarising the full regret-matrix, each climate is characterised by—first, the strategy that has the lowest worst-case regret (mini-max criterion[41]) across all the assessed climates. Second, the strategy that had the lowest 75th percentile regret across all climate futures (Domain criterion[42]).

The mini-max criterion assumes a high-risk aversion approach to selecting a strategy to avoid extreme adverse outcomes, but there is a possibility that the chosen strategy is disproportionately influenced by extreme cases. The second approach (a variant of the domain criterion) is less sensitive to the extremes. This methodology, called Robust Decision Making[43] has been successfully applied in similar settings related to water and wastewater management, and flood risk assessments to support decision making under high levels of uncertainty[44–46].

**Code availability**. The OSeMOSYS code, used to develop the EAPP energy model, is available on a GitHub repository.

## Data availability

The base data used to construct the EAPP energy model is available as part of the supplementary material (Supplementary Tables 5–15). A model file, of the EAPP's electricity system, developed for this study has been deposited in the following Zenodo data repository[47]. All relevant source data for the figures in the main article and the supplementary material are also available in the same repository.

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

## Acknowledgements

This work was primarily supported by the World Bank (contract number 8004717) and their funders—Agence Française de Développement (Afd). We would also like to thank the Swedish research councils— Vetenskapsrådet and Formas (grant number 942-2015-1304)—for their support.

## Author contributions

V.S. developed the energy models described in this article. V.S., O.B., A.S. and B.B. performed the climate scenario iterations and contributed to the overall methodology. M.H., H.-H.R. and C.T. provided valuable inputs to develop the energy model. R.C., J.E.N., K.M.S., M.H., B.B., R.L., D.G.G. developed the methodology of the study. B.B. performed the bias-correction and downscaling of climate scenarios. R.L. and D.G.G. developed the robust decision-making analysis (RDM) component of the study. B.J. and A.H.-L. developed the water models for this study and contributed to the overall methodology. V.S., O.B., A.S., M.H., C.T., H.-H.R., B.B. and D.G.G. were involved in paper writing and editing.

## Additional information

**Competing interests:** The authors declare no competing interests.

