## [Peer Review File · Nature Communications]

Reviewers' comments:

Reviewer #1 (Remarks to the Author):

Review "Resilience of Eastern African Electricity Supply Infrastructure to Climate Change"

This paper presents a modelling framework to assess EAPP's expansion plans taking into account hydro-climatic uncertainties. The framework involves the loose-coupling of a least-cost investment model covering the EAPP with a water system model of the Nile River Basin. This methodology is not new but its implementation in the EAPP/Nile region is.

I have several concerns with this paper:

- We need more information on both models, and more specifically about their implementation (planning horizon, time step, input data, etc.). For example, in the Method section, we can read that the EEAP model has a temporal resolution of 48 months while WEAP is run over 20 years on a monthly time step. How do you manage the mismatch between the temporal resolution in both models? Is the planning horizon the same for both models?
- When a large hydro investment is made, how does WEAP handle the filling of the reservoir? This is a controversial issue in the Nile, especially for upstream reservoirs.
- Main Text. I disagree with the statement "Amongst the CMI values for the illustrated African basins, high level of uncertainty is expected in the Nile River Basin". The Eastern Nile, which contributes to about 80% of the Nile, displays less uncertainty than the majority of the rivers listed on Fig 1.
- The authors claim "the modelling framework assesses power pool infrastructure resilience to the effect of a changing " but could the framework be also used to assess water users' resilience?
- The modelling framework only handles the uncertainty of the supply-side. What would it take to extend the framework so that it can also handle the load uncertainty? Considering population growth rates in the region and the uncertainty regarding access to electricity, the load uncertainty is likely to be significant.
- Page 6. What does the term "consumer" mean here? I guess Egyptian consumers are actually players on the bulk electricity market (industries, utilities, etc.) not end-users like households.
- Why are the fluctuations in electricity cost more noticeable for the no adaptation scenario?

- Why did the authors choose an implementation based on the continuous simulation under a non-stationary climate instead of multiple simulations of a distant, stationary, future? This second implementation is often adopted by hydrologists to assess the impact of climate change on flow regime. It allows more robust comparisons with historical conditions. In other words, can we really conclude that the cost of electricity in, say Uganda in 2031 under the no-adaptation will be around 0.035 \$/kWh, a 0.015\$ increase with respect to 2028 (Fig 4-a)?!? Presenting such results is misleading without proper explanation as to how to interpret the figure.
- Why does the consumer expenditure change so much in Uganda in the most resilient strategy? In less than 10 years, we can see that the generation cost is reduced by half without a significant change in the supply mix (Fig 3). Where does that change come from?
- The main conclusion is evident: “Our results indicate that the most resilient strategy for the countries in the EAPP is expected to be one planned for a climate that is cumulatively wetter compared to the baseline (supplementary section 2.2); the strategy had the lowest value for the maximum regret across the different climate futures. Why would we expect a different result?
- The EEAP model seems to minimize investment plus generation costs over the planning horizon given a hydro sequence corresponding to a particular climate scenario. The investments are then imposed on to WEAP, which then allocates water between competing uses. Since hydropower generation depends on the reservoir operating policies, (1) can the operating policies in WEAP change with time/investment? (2) is there a feedback loop from WEAP to the EEAP model (e.g. reduced water availability due to irrigation withdrawals), (3) could probabilistic states be included in the EEAP model to better reflect the uncertainty regarding the availability of hydro units?

Reviewer #2 (Remarks to the Author):

The paper investigates the impact of climate change on the resilience of the electricity supply infrastructure in the Nile River Basin. In particular, the authors suggest that current energy plans don't consider the impact of climate change on water, which will affect hydropower potential. The authors therefore consider the impact of three types of model runs; i) based on historical climate, ii) adaptation with perfect foresight of future climate (PF) and, iii) no-adaptation (NA).

The paper is reader-friendly. Relevant information that is missing in the main text is provided in the supplementary material. This is important because it means the results sound replicable. Furthermore, open-source models are used, which is another good point.

I'm not an expert on African issues, but according to my knowledge on other regions, the paper's results are original and interesting.

I would like to highlight three points, which should be further discussed from my perspective:

1) The authors consider three "types of model runs" i.e., baseline, PF, and NA. Unless I missed the point, the most comprehensive approach would have been to consider adaptation, assuming the decision-makers don't know the specific future climate scenario that will occur. It would represent the best strategy. I guess it is quite hard to consider this option because of computational burden. It is probably out of the scope of the paper. Nevertheless, I would recommend mentioning the reason why you don't consider this "type of model runs" and the implication regarding the related limitation.

2) My second concern is there are drivers with a higher level of uncertainty than climate change. On page 10, there is reference to the fact that not all sources of uncertainty are investigated, and some are mentioned. Nevertheless, I'm wondering if the author shouldn't further discuss that point. For instance, does it make sense to focus on climate change when the model outputs are probably far more affected by the technology cost uncertainty? What is the validity of the conclusions in view of the existing uncertainties? I agree one can't take everything into account however I believe that these points should be addressed further as opposed to merely mentioned.

3) Partially linked to this point, Table 12 of the supplementary document provides the cost of technology. I'm wondering if the authors consider the cost evolution (learning curve). This is a driver that may significantly affect the outcome. For instance, the cost of photovoltaic decreases more than the cost of hydropower. If I'm right, there are four scenarios developed in the quoted reference (IRENA). Do you consider a specific one? I should confess that I haven't found this information in the paper.

A last minor remark concerns Fig 3: I would recommend defining the acronym. I guess the HFO means "heavy fuel oil" and ACOE "average cost of electricity", but it would be better to provide this information.

The investigation sounds original and rigorous. I do however feel that the above-mentioned points should be improved in order to better justify certain results. I recommend the publication of the paper after minor revisions.

Don't hesitate to contact me if you need further information.

Best regards,

Ludovic Gaudard

Reviewer #3 (Remarks to the Author):

Overview

The paper addresses an interesting and timely topic and provides an interesting result.

The main result is that in East African countries where a power sector capacity expansion model keeps a substantial amount of hydropower in the generation mix through to 2050, electricity generation costs would fluctuate substantially between wet and dry years. Considering deployment strategies that are robust to different climate conditions reduces this variability and keeps costs lower. This is what one would intuitively expect but it is good to see it quantified.

These results should be of interest to others in the field of energy planning and energy system modelling.

As details of the code and data are not made available, I cannot comment in detail on the modelling method beyond stating that it looks reasonable overall if my concerns given below can be addressed. The water system model is outside my area of expertise and I cannot assess its suitability for the use it has been put to in this study.

I have some concerns with the methods, and think that the paper needs major revisions to address these concerns and to provide more information on methods and results, as outlined below.

Major comments

1. Novelty and link to World Bank report

I don't understand the last paragraph on p.7. You write "Our results indicate that the most resilient strategy for the countries in the EAPP is expected to be one planned for a climate that is cumulatively wetter compared to the baseline". That sounds like a result. But you then say that this strategy includes adaptation in the water and agriculture sectors, which this paper does not discuss, and refer to a World Bank report: "Full results from the study including other performance indicators are available in Cervigini et al.(4)". So are you saying that the paper presents some of the results from the World Bank report but without fully explaining the metrics used (i.e., this paper talks about the energy sector only, but in determining the most resilient strategy you also considered other sectors)? This needs further clarification - also in light of the novelty criterium for Nature Communications.

2. Capacity expansion modelling

As far as I understand you are using expansion plans set by the different countries through to 2050 as minimum capacity constraints for your generation expansion planning, and allow your model to add additional capacity if necessary. In Figure 3 you show that by 2050 Tanzania goes from all hydropower to about 30% hydro with the majority of the remainder coming from coal generation. According to table 12 in the appendix the capital cost of CSP, PV and wind technologies is lower than that of coal. Leaving aside the question of how you represent the variability of wind and solar generation in your model, this would suggest that your cost-minimizing model should build a lot of wind and solar generation. It seems strange to me that only coal and gas is deployed, with no PV at all visible in Figure 3 for Tanzania - unless this what PIDA+ says for Tanzania.

In any case, you need to make it clearer in the paper what the PIDA+ plans are for each country and what your model says (and to what extent they differ). I would expect figures similar to Figure 3 that show what PIDA+ says for each country, or for each of these figures to show the difference between PIDA+ and your model results.

The treatment of weather-dependent variability of renewable generation and demand in the capacity extension model is not explained sufficiently. Past work (e.g. Bloomfield et al., doi: 10.1088/1748-9326/11/12/124025) has shown how important this variability is for the results of power system models irrespective of future climate variability. How does the time slice approach of 12 months and 2 day-types consider weather-driven inter-annual variability of electricity demand and of supply from variable renewable generation technologies like PV and wind?

In Figure 4, the no-adaptation electricity generation costs for Tanzania and Uganda fluctuate quite a bit through time. This suggests to me that there in addition to the climate signal there is substantial year-by-year variability in the data that the authors do not address, but which could have a strong influence on results.

It is also not clear to me why costs should increase through time. Given the falling costs of major technologies it would seem more sensible that generation costs fall with time.

3. Scenario results and selection of scenarios to compare in the paper

As far as I understand, the EAPP electricity sector planning model is run for the 6 selected climate realizations (perfect foresight model runs), then the model is run with each of the 6 resulting optimal expansion plans but using climate data from the 5 other scenarios. In addition the no-adaptation expansion plan (optimized for current climate) is also compared to the 6 future climate scenarios, resulting in a total of 42 combinations of electricity sector expansion plans and climate conditions. From these, 2 sets are chosen for each country: the most resilient adaptation strategy of the 6 planning strategies considered, and the no-adaptation expansion operated under the driest climate conditions of the 6 future climate scenarios. Is this interpretation correct? I think it would be nice to have a table somewhere explaining these different scenarios. Little concrete detail is given on the two criteria used to select the most resilient adaptation strategy. These should be described in more detail. See also the comment above on how this study relates to the World Bank report cited as ref number 4. If the authors do not intend to release the full results, they could show figures or tables of results of the perfect foresight capacity expansion model runs for one or two countries to show the range of capacities that result in different electricity generation technologies.

4. Uncertainty of other factors vs uncertainty in climate

One of the key results seems to be on p.7: "Uganda and Tanzania could expect costs to vary could expect costs to vary between -18% and +28%, -5% and +23% respectively between the wettest and driest climates when compared to the baseline" (the authors should clarify though whether this statistic is for 2050 or across some average of years). While these are substantial numbers, they do beg the question what the variability introduced by other factors would look like.

The authors state in the SI, last para of section 1.2.1, that because of the considerable uncertainty in projecting energy demand "the results need to be analysed with a grain of salt." However, I do not

see this uncertainty really being dealt with in the study. There seems to be a single demand projection for each country in the EAPP. For example, electricity demand in Tanzania is projected to grow from 7.5 TWh in 2015 to 69.73 TWh in 2050 according to Table 5 in the SI appendix. How would the results of this study change if demand were only half as high, or double that value, in 2050? The possible uncertainty seems enormous and because the paper does not present much detail on the different sets of electricity generation technologies deployed under the different climate scenarios, the reader cannot judge whether the uncertainty introduced by demand (i.e., things like economic and technology development) would dwarf the uncertainty introduced by climate through 2050. In particular, given that in the scenarios presented here the vast majority of electricity demand across EAPP is satisfied by gas and coal generation in 2050, coal and especially gas prices would seem like such a major source of price volatility that they might also dwarf the effect of different climates. How, if at all, does the study consider this?

The second para on p.10 states: "These results emphasise the message that irrespective of the climate, availability of electricity trading infrastructure stands to play a significant role in balancing electricity prices within the EAPP." The authors could expand on this to provide a better overview of how large an effect climate uncertainty has on the other major sources of uncertainty in the development of the EAPP economies and their power systems.

Specific comments

The paper would benefit from a thorough proofreading to improve the quality of English.

Introductory paragraph: It seems strange to have only a single reference, and that reference to be to the Platts database. I would expect that to be to a definition of EAPP.

Footnotes are not used in Nature journals as far as I know, but in any case, they should not use the same numbering and style as endnotes - the two are currently indistinguishable in the text. For example, on p.5 you write "... the PIDA+4,23 expansion plan" where 4 seems to refer to a footnote and 23 to an endnote. You should explain what PIDA+ is in directly in the main text.

Figure 1 label is unclear. I suggest something like: "Projections of mean annual CMI across Africa's major river basins under different radiative forcing scenarios. The CMI (Climate Moisture Index) is a measure of aridity [...]"

Para 2 on p.5 states that the EAPP power system expansion model contains "all individual hydropower plants, which were represented in the water model". A map of these plants would be nice to see, but more importantly, some summary statistics on the variability of electricity generation of these different plants under the different modeled climate future scenarios. That would help the reader better understand the value of considering each individual hydro plant in this study, i.e. whether the modelling methods used are actually able to actually represent local variability in hydro generation.

Figure 3 needs a better color scheme. It is almost impossible to tell the colors for diesel and geothermal apart. Perhaps 5-year increments rather than plotting each individual year would make the figure less busy and easier to read? Also, the figure caption is not clear. Do all four panels show the cost-optimal expansion plans (for either EAPP or the three individual countries) under the baseline (historical) climate? Why were these three countries selected?

Can you add the country-specific results analogous to Figure 3 for the remaining countries as supplementary figures? It would be interesting to see what the expansion scenarios look like for deployment optimisation runs other than the baseline.

Figure 4 needs a better caption. You should describe all three panels in the caption so that the figure can be understood without searching for where it is described in the text. Perhaps here you could also include the deployed generation technology mix (maybe in 5 year increments to keep the figure readable) for comparison?

Figure 5 needs a more detailed label to explain the difference between worst case and most resilient strategy.

Figure 6: Visually interesting, but I don't see any explanation of the bar chart alongside the outer circumference of the circles. What are these percentage values showing?

Methods

First para of methods on p.11 has two consecutive sentences starting with "The EAPP model " - I think the second of these should be "The WEAP model" (maybe a bad idea to give these models such

similar names? It certainly makes reading the manuscript an exercise in concentration to make sure one does not mix up these two).

You state that over 500 electricity generation technologies were included, can you clarify what these >500 technologies are?

Is technological learning considered at all in the model? It would seem very likely that the cost of PV and in particular of batteries continue falling substantially and that by 2050 the relative cost difference between PV and e.g. coal will look very different than what table 12 now states (see e.g. Schmidt et al, <http://dx.doi.org/10.1038/nenergy.2017.110>).

It seems unrealistic to assume that only currently planned transmission capacity will exist in 2050. Can this assumption be justified? How different would model results look if the planning model were allowed to increase the transmission capacities between countries?

Can you clarify the reason for also including older CMIP3 data as inputs to the WEAP model? To me it would make a lot more sense to limit yourself to CMIP5, instead consider more than two RCPs, and possibly show data split by RCP rather than between CMIP3 and CMIP5 (in Figure 1).

The authors only consider electricity demand and supply. How do other energy-using sectors factor in to the analysis, for example mobility?

Supplementary material: Referencing the SI would be easier if all items were numbered consecutively with Sxxx, for example, Table S11 rather than Annex A1-Table 11. Also the SI should have page numbers.

Appendix Table 11 has no source.

Tables 6 and 7 in the supplementary appendix are unclear. What are the four rows labelled Part 1 - Part 4?

Resilience of Eastern African Electricity Supply Infrastructure to Climate Change

Response to Reviewers

The authors wish to thank the reviewers for their feedback that helped improve the quality of the paper. The comments from the reviewers and the corresponding responses and actions taken are provided below.

Reviewer #1

Review “Resilience of Eastern African Electricity Supply Infrastructure to Climate Change”

This paper presents a modelling framework to assess EAPP’s expansion plans taking into account hydro-climatic uncertainties. The framework involves the loose-coupling of a least-cost investment model covering the EAPP with a water system model of the Nile River Basin. This methodology is not new but its implementation in the EAPP/Nile region is.

Comment from Reviewer #1	Response-Action
We need more information on both models, and more specifically about their implementation (planning horizon, time step, input data, etc.). For example, in the Method section, we can read that the EEAP model has a temporal resolution of 48 months while WEAP is run over 20 years on a monthly time step. How do you manage the mismatch between the temporal resolution in both models? Is the planning horizon the same for both models?	Thank you for pointing out the need to more clearly describe the temporal resolution description – this has thus been improved in the methods section and the supplementary material. In summary: the long-term expansion-planning model of the East African Power Pool (EAPP) has 48 splits in each modelled year; each year is divided into 12 months and each month is further classified into 4 parts. (As the emphasis was on examining seasonal fluctuations at a monthly level, daily descriptions were kept coarse). The splitting of the time steps is expounded as part of the supplementary material (section 1.2.2). The WEAP model of the Nile Basin, on the other hand, has a monthly time step i.e. each year in the WEAP model is represented by 12 splits corresponding to each month. The modelling/planning horizon for both the models is the same, and it extends until 2050. To manage the difference in temporal resolutions, we decided to match the monthly time step of the water model with the electricity sector model. The water model produced climate-specific hydropower generation estimates on a monthly time resolution (as monthly capacity factors); the electricity sector model was set up such that all the four daily splits in each month had the same capacity factors as the water model. The capacity factor calculation and the model integration are explained in the supplementary material (section 1.5). The input data (power plant capacities, fuel prices, fossil fuel reserves, technology and country-specific costs etc.) for the electricity sector model are

Comment from Reviewer #1	Response-Action
	included as part of the supplementary material (Appendix). Basic assumptions in the WEAP model are specified in the supplementary material (section 1.1). (With respect to our assertion that WEAP has been run over the '20 years'. We intended to communicate that WEAP has been used and refined over the past 20 years. The comment was not related to the model period. The ambiguity has been addressed in the paper.)
When a large hydro investment is made, how does WEAP handle the filling of the reservoir? This is a controversial issue in the Nile, especially for upstream reservoirs.	When investments are made in large hydropower plants, the reservoir nodes in WEAP can simulate the filling up of the reservoirs (in the case of dammed hydro plants). WEAP has an option to specify an 'initial storage level' and a 'volume elevation curve' to estimate the time required for the reservoir to fill up, taking into consideration the flow (m³/s) of the river draining into the reservoir. For new hydropower plants in the Nile River Basin, damming of water in the upstream countries has resulted in geopolitical debates due to potential non-availability of water for downstream countries, as pointed out by the reviewer. To avoid any misrepresentation in the model, reservoir characteristics, hydropower plant configuration and initial reservoir storage levels were obtained from discussions with local contact points at the respective sub-basin committees - and through stakeholder workshops involving organizations that contribute to the hydropower sector. A reservoir in WEAP has four different zones: Flood control zone, conservation zone, buffer zone and an inactive Zone. Each reservoir in the Nile Basin WEAP model has its zone definition and release priority based on other water demands in the region. The modelled hydrology then determined the simulation of the dam filling process.
Main Text. I disagree with the statement "Amongst the CMI values for the illustrated African basins, high level of uncertainty is expected in the Nile River Basin". The Eastern Nile, which contributes to about 80% of the Nile, displays less uncertainty than the majority of the rivers listed on Fig 1.	We agree with the reviewer's remarks on climate uncertainty, based on the CMI values for the different African Basins (figure 1). Our phrasing was clumsy and has been corrected. Historically, the Eastern Nile River Basin (which contributes to about 80% of the annual flow on the Nile River) has been dry and almost all the analysed climate futures indicate a similar trend to continue. On the other hand, the CMI values (from CMIP 5 models) indicate that the White Nile (or Nile Equatorial lakes) region, indeed indicate a high uncertainty amongst the analysed African River Basins and is expected to be wetter than the Blue Nile projections. Despite the White Nile contributing to approximately (only) 20% of total annual Nile River Flow at Khartoum, it provides almost all the flow for ~4 months in a year; this makes the uncertainty in the White Nile Basin substantial enough to analyse. Moreover, from an energy system perspective, there is approximately 2000 MW of future hydropower investment in Uganda, which solely depend on the White Nile. We have modified the text in the main paper to reflect the thoughts mentioned above.

Comment from Reviewer #1	Response-Action
The authors claim “the modelling framework assesses power pool infrastructure resilience to the effect of a changing “but could the framework be also used to assess water users’ resilience?”	The framework could indeed be used to assess selected aspects of water users’ resilience. It considers infrastructure expansion plans in the energy, agriculture, and water use sectors; in addition to capturing the increasing household and industrial water demands across the Nile River Basin. Water consumption in the agricultural sector covers seasonal water demands based on the type of crops grown in a particular Sub-Basin. Hence, the water stress that could potentially arise from any of the analysed climate futures could be calculated. Those stresses could then be evaluated (using different allocation priorities) to determine the vulnerability of the users. And, with defined user vulnerability, the efficacy of specific resilience measures could be evaluated. That is beyond the scope of this work. However, a recommendation to consider this potentially important extension is made.
The modelling framework only handles the uncertainty of the supply-side. What would it take to extend the framework so that it can also handle the load uncertainty? Considering population growth rates in the region and the uncertainty regarding access to electricity, the load uncertainty is likely to be significant.	The additions required to consider the uncertainties related to electricity demand (load) and other variations on the demand side are minimal. We hope that by demonstrating the use of this approach for a new (though limited) application, we can set the groundwork for future research – and extend this framework further. While we deliberately chose to keep much constant in this piece to help with tractability, there are a number of important uncertainties that can be included in future analysis. Those of particular interest include fuel, efficiency, demand, trade, emissions constraints and technology development. As with the prevision comment, a recommendation to consider this potentially important extension is made.
Page 6. What does the term “consumer” mean here? I guess Egyptian consumers are actually players on the bulk electricity market (industries, utilities, etc.) not end-users like households.	The “consumer” in the term consumer-expenditure used in this study, refers to all the electricity consumers within each of the analysed countries. The authors agree with the reviewer’s remark on different categories of electricity consumers (and as this contribution is built upon, should be delineated). However, in this study, a deeper classification of end users is not the focus, as we do not look in this round to demand side resilience measures. For simplicity, we assume that a ‘cost reflective’ tariff is applied – and the full cost is recovered across the consumers. (This avoids, among other things the need to account for complex ‘cross subsidies’ that occur as a more detailed representation is included). This single consumer representation, though not ideal, provides a balanced (and consistent) metric for assessing the climatic impact on the power infrastructure.

Comment from Reviewer #1	Response-Action
Why are the fluctuations in electricity cost more noticeable for the no adaptation scenario?	In the no-adaptation scenario, the rollout of large energy infrastructure is fixed as per the perfect foresight strategy (PF) of a climate future. Shortfalls in generation due to the lack of water (climate-induced) for hydropower were allowed to be satisfied only with rapidly deployable, yet expensive, diesel-based generation options. For example, if three climates are analysed (A, B, C), the infrastructure rollout PF strategy for climate A was fixed, and two model runs were performed with the other climates (B & C). In these two model runs, expensive fossil fuel based generation options were used to compensate for the loss of hydropower generation, if any. Since the cost of diesel is high (country specific and obtained from the latest EAPP master plan), its effects are realised on the cost of generation, and hence the fluctuations are pronounced. (In reality, the economic damage may be higher – were diesel generators not deployed in time and production in some industry were halted etc.). This is pronounced for countries with high hydro-power deployment such as Uganda and Tanzania. Conversely, countries with a lower share of hydropower generation, such as Egypt, the NA scenario leads to dampened fluctuations in cost.
Why did the authors choose an implementation based on the continuous simulation under a non-stationary climate instead of multiple simulations of a distant, stationary, future? This second implementation is often adopted by hydrologists to assess the impact of climate change on flow regime. It allows more robust comparisons with historical conditions. In other words, can we really conclude that the cost of electricity in, say Uganda in 2031 under the no-adaptation will be around 0.035 \$/kWh, a 0.015\$ increase with respect to 2028 (Fig 4-a)?!? Presenting such results is misleading without proper explanation as to how to interpret the figure.	The “continuous simulation” approach was employed because of the need to present a consistent and sensible investment trajectory from the present through 2050. Although the authors have employed the approach the reviewer recommends in numerous other studies (sometimes called the “era” approach), these typically involve analyses of crop yields, river runoff, drought occurrence, or other variables that do not have significant year-to-year dependencies (e.g., a hydropower investment in 2020 affects decisions in 2030). Making multiple simulations of a stationary future, say 2040-2050, would also require making assumptions about initial conditions in 2040, which introduces significant uncertainties. We recognize the difficulties the simulation approach creates in making comparisons to the baseline, but the primary aim of this study is to evaluate the benefits of resilient adaptation to a “no adaptation” future, rather than the historical baseline. Regarding the cost of electricity generation, which is used as a metric to discuss results, generally, the costs will be higher if the climate is dry and the electricity system is not properly adapted (especially when upstream water demands are given preference). In the case of Uganda, the cost of generation is 90% higher in 2031 in comparison to 2028. This is because, in 2031, due to its high dependence on hydropower coinciding with an extreme dry year, the system is forced to use expensive fossil fuel options to meet the demand. The costs discussed are an indication of the level

Comment from Reviewer #1	Response-Action
	of impact of no-adaptation. Thank you for pointing out that this was potentially misleading. The text has been updated.
Why does the consumer expenditure change so much in Uganda in the most resilient strategy? In less than 10 years, we can see that the generation cost is reduced by half without a significant change in the supply mix (Fig 3). Where does that change come from?	Figure 3 shows the generation mix and the consumer electricity expenditure in selected countries of the Eastern African Power Pool (EAPP) for the baseline strategy and not the most resilient strategy. In 2017, the generation mix in Uganda has hydro, geothermal and some generation from distributed diesel generators. By 2027, the share of hydropower in the total generation mix increases by 5 % in the generation mix, replacing almost all the distributed diesel-based generation. Hence the drop in expenditure. Diesel fuel has a high variable cost and their replacement creates a drop in the annualised cost of electricity generation. We can also notice that later in the modelling period when the fossil fuel usage increases, the expenditure starts to rise; in this case due to the usage of locally available Natural gas.
The main conclusion is evident: “Our results indicate that the most resilient strategy for the countries in the EAPP is expected to be one planned for a climate that is cumulatively wetter compared to the baseline (supplementary section 2.2); the strategy had the lowest value for the maximum regret across the different climate futures. Why would we expect a different result?	Again, many thanks for the comment. We have updated the text to read “an important insight of the analysis is that the most resilient strategy is one planned for a wetter climate; this would not have been intuitively expected prior to this analysis. One might have anticipated that variations would lead one to simply move from Hydro.” We feel that this would not have been immediately obvious.
The EAPP model seems to minimize investment plus generation costs over the planning horizon given a hydro sequence corresponding to a particular climate scenario. The investments are then imposed on to WEAP, which then allocates water between competing uses. Since hydropower generation depends on the reservoir operating policies, (1) can the operating policies in WEAP change with time/investment? (2) is there a feedback loop from WEAP to	Again, many thanks for the comment. The corresponding text in the paper has been sharpened. Prior to the start of the model runs, potential start dates are synchronised between the water (WEAP) and energy (OSeMOSYS) models with information obtained from PIDA+. The information flow is set in such a way that the water model (WEAP) is run first and the maximum hydropower generation is communicated to the energy model (as a capacity factors). Then, the energy model is run, and the start dates and capacities of hydropower plants from the optimization process are fed back into the WEAP model for water allocation (in the second iteration). There exists a feedback loop (soft-linking) in the framework between the two sets of models and they are iterated twice for concurrency.  1. The operating policies (priority of water allocation to the different uses: household consumption, agriculture, hydropower) in WEAP are set from the beginning. What changes

Comment from Reviewer #1	Response-Action
the EEAP model (e.g. reduced water availability due to irrigation withdrawals), (3) could probabilistic states be included in the EEAP model to better reflect the uncertainty regarding the availability of hydro units?	is the hydropower plant capacity and the year in which it can start generating electricity (which is decided by the OSeMOSYS optimization process). In parallel, the demand for water in agriculture and households increase depending on parameters like population increase, water consumption per capita and irrigation plans in respective countries. In the second iteration, depending on the hydropower plant capacity suggested by the EAPP model, the WEAP model produces a new estimate for max hydropower generation, taking into consideration the different demand priorities and climate.  2. Yes, there exists a feedback loop in the framework. We have two iterations starting from the WEAP model in the first. As mentioned in point 1, periodic irrigation water withdrawals are part of the WEAP model which considers different irrigation plans in the constituent countries of the power pool. 3. OSeMOSYS is a deterministic, least cost based linear optimization system; hence, the probabilistic states cannot be included. That being said, the framework has the flexibility to include different hydro units as available options for investment and can be chosen if it leads to an, over all, least system cost.

Reviewer #2 (Remarks to the Author)

The paper investigates the impact of climate change on the resilience of the electricity supply infrastructure in the Nile River Basin. In particular, the authors suggest that current energy plans don't consider the impact of climate change on water, which will affect hydropower potential. The authors therefore consider the impact of three types of model runs; i) based on historical climate, ii) adaptation with perfect foresight of future climate (PF) and, iii) no-adaptation (NA).

The paper is reader-friendly. Relevant information that is missing in the main text is provided in the supplementary material. This is important because it means the results sound replicable. Furthermore, open-source models are used, which is another good point.

I'm not an expert on African issues, but according to my knowledge on other regions, the paper's results are original and interesting. I would like to highlight three points, which should be further discussed from my perspective:

Comment from Reviewer #2	Response-Action
The authors consider three "types of model runs" i.e., baseline, PF, and NA. Unless I missed the point, the most comprehensive approach would have been to consider adaptation, assuming the decision-makers don't know the specific future climate scenario that will occur. It would represent the best strategy. I guess it is quite hard to consider this option because of computational burden. It is probably out of the scope of the paper. Nevertheless, I would recommend mentioning the reason why you don't consider this "type of model runs" and the implication regarding the related limitation.	We thank the reviewer for pointing this out. We note that each of the perfect foresight (PF) strategies are actually the perfect adaptation strategy for their respective climates as the model sees them coming – with perfect foresight. (i.e. it is the 'perfect response' to the change in climate that is programmed into the model.) This represents an extreme. It is the best that could possibly be done, as the changes are perfectly anticipated. As pointed out by the reviewer, however, in reality, it is not clear how the unexpected changes would occur. Thus, any strategy taken (no matter how good) would be a little less than ideal. Similarly, the no-adaption scenario is an extreme case. It is what happens if almost nothing were done to change the system onto which an unexpected climate is forced. Reality will be somewhere between – and the two PF and NA scenarios show us the envelope in which we are likely to operate. From these, we can draw insights. We choose a set of representative climates (the explanation for why those, is provided in section 1.3 of the supplementary material. The insights drawn are first of a kind for the region and useful, but limited. If it were the objective to design a strategy for the EAPP in the face of uncertainty - we would test a much larger ensemble of infrastructure configurations. These would be assessed against potential future climates. We would also include other uncertainties (noted above). As requested, we draw the readers' attention to this limitation.
My second concern is there are drivers with a higher level of uncertainty than climate change. On page 10, there is reference to the fact that that not all sources of uncertainty are investigated, and some are mentioned. Nevertheless, I'm wondering if the author shouldn't further discuss that point. For instance, does it make sense to	We agree to the reviewer's concerns about uncertainties other than climate change. As mentioned on page 10, we addressed only the uncertainty of climate change in this study. We have expanded the discussion on page 10 to discuss the non-climate related uncertainties in greater depth and their potential impact on results.

focus on climate change when the model outputs are probably far more affected by the technology cost uncertainty? What is the validity of the conclusions in view of the existing uncertainties? I agree one can't take everything into account however I believe that these points should be addressed further as opposed to merely mentioned.	
Partially linked to this point, Table 12 of the supplementary document provides the cost of technology. I'm wondering if the authors consider the cost evolution (learning curve). This is a driver that may significantly affect the outcome. For instance, the cost of photovoltaic decreases more than the cost of hydropower. If I'm right, there are four scenarios developed in the quoted reference (IRENA). Do you consider a specific one? I should confess that I haven't found this information in the paper.	Table 12 & 13 in the supplementary section provide the capital costs for all the generic and site-specific power plants that could be part of the power system. In this study, for all the generic power plants, we consider a capital cost that does not change over time in the modelling system. The values correspond to the technology-specific starting value in all the four different scenarios mentioned in IRENA's analyses of the Southern African Power Pool (SAPP) (IRENA, 2013). We agree with the reviewer's opinion that the capital cost of Solar PV is expected to decrease and would be cost competitive with the hydropower-based generation. However, according to the renewable promotion scenario in the aforementioned IRENA study (where a technology learning curve is applied, and the capital cost of non-hydro renewables decrease over time), even by 2030, the cost (levelised cost of electricity generation (LCOE)) of grid-connected PV is expected to be 39% more than that of hydro-based electricity generation; the share for rooftop PV is expected to be 56% more in 2030. We agree that this share is expected to decrease and by 2040, it could compete with hydropower and this is something to be taken into consideration while drawing conclusions from the results of the modelling period. This model does not consider the increase in integration costs that could realise from very high penetration of renewables (wind and solar), which might significantly affect the optimal mix. We have added a note to the main document addressing this issue. That being said, considering that most of the hydropower investment is expected to occur from now leading up to 2035-2040, it remains interesting to observe how much will be the share of distributed and grid-connected PV in future.
A last minor remark concerns Fig 3: I would recommend defining the acronym. I guess the HFO means "heavy fuel oil" and ACOE "average cost of electricity", but it would be better to provide this information.	Thank you for pointing this oversight out. We have corrected the acronym throughout.

The investigation sounds original and rigorous. I do however feel that the above-mentioned points should be improved in order to better justify certain results. I recommend the publication of the paper after minor revisions. Don't hesitate to contact me if you need further information.

Reviewer #3 (Remarks to the Author):

Overview

The paper addresses an interesting and timely topic and provides an interesting result.

The main result is that in East African countries where a power sector capacity expansion model keeps a substantial amount of hydropower in the generation mix through to 2050, electricity generation costs would fluctuate substantially between wet and dry years. Considering deployment strategies that are robust to different climate conditions reduces this variability and keeps costs lower. This is what one would intuitively expect but it is good to see it quantified. These results should be of interest to others in the field of energy planning and energy system modelling. As details of the code and data are not made available, I cannot comment in detail on the modelling method beyond stating that it looks reasonable overall if my concerns given below can be addressed. The water system model is outside my area of expertise and I cannot assess its suitability for the use it has been put to in this study. I have some concerns with the methods, and think that the paper needs major revisions to address these concerns and to provide more information on methods and results, as outlined below.

Response: We thank the reviewer for his detailed comments and suggestions. We have responded to his comments in the following sections. With respect to the code that was used for this analysis, it can be accessed at this GitHub link. There is a long and a short version of the OSeMOSYS code. We started with the long one and ended up shortening it to improve optimization times.

Major comments

Comment from Reviewer #3	Response-Action
1. Novelty and link to World Bank report 1.1 I don't understand the last paragraph on p.7. You write "Our results indicate that the most resilient strategy for the countries in the EAPP is expected to be one planned for a climate that is cumulatively wetter compared to the baseline". That sounds like a result. But you then say that this strategy includes adaptation in the water and agriculture sectors, which this paper does not discuss, and refer to a World Bank report: "Full results from the study including other performance indicators are available in Cervigini et al.(4)". So are you saying that the paper presents some of the results from the World Bank report but without fully explaining the metrics used (i.e., this paper talks about the energy sector only, but in determining the most resilient strategy you also considered other sectors)? This needs further clarification - also in light of the novelty criterium for Nature Communications.	The metrics and results in this article (country-specific results, trade analysis, and regret matrix calculations) are not reported elsewhere. Figures 1 and 5 (main article), are an exception, adapted from (Cervigni et al., 2016). The work presented in this article is part of a two-year project commissioned by the World Bank Group. A number of other results – as well as complementary performance indicators – can be found in (Cervigni et al., 2016). The reviewer’s understanding is correct; this article focusses only on the power system vulnerability. We discuss the power sector and its climatic impact in sufficient detail and also give a brief overview of the WEAP model (water and agriculture sectors) in the supplementary section, but do not report the water balance modelling in detail here.
2. Capacity expansion modelling 2.1 As far as I understand you are using expansion plans set by the different countries through to 2050 as minimum capacity constraints for your generation expansion planning, and allow your model to add additional capacity if necessary. In Figure 3 you show that by 2050 Tanzania goes from all hydropower to about 30% hydro with the majority of the remainder coming from coal generation. According to table 12 in the appendix, the capital cost of CSP, PV and wind technologies is lower than that of coal. Leaving aside the question of how you represent the variability of wind and solar generation in your model, this	2.1 We would like to clarify that the expansion plans in all countries are only committed to building plants that have been committed to operate with a start date clearly defined. Our model allows for new investment limited to the set of technologies identified by the PIDA+ studies. The reviewer correctly points to the fact that in our results lower capital cost does not drive up the adoption of solar or wind technologies. The model bases its choice of investment on the lowest NPV cost of the system. Thus, it accounts for life-times, capacity factors, total annual electricity generation, variable and fixed costs and fuel costs, in addition to the capital cost. The capacity factors in this case are important – and account for the

would suggest that your cost-minimizing model should build a lot of wind and solar generation. It seems strange to me that only coal and gas is deployed, with no PV at all visible in Figure 3 for Tanzania - unless this what PIDA+ says for Tanzania. In any case, you need to make it clearer in the paper what the PIDA+ plans are for each country and what your model says (and to what extent they differ). I would expect figures similar to Figure 3 that show what PIDA+ says for each country, or for each of these figures to show the difference between PIDA+ and your model results.	observation. A capacity factor of 25% for wind turbines would imply multiplying its capital costs to allow it on average to produce the same as a coal power plant of the same size (running at ~95%). Thus, though wind's capital cost is low, more capacity needs to be installed – and more money spent – thus it would drive up the NPV and is not invested in. Further, the capacity factor for solar (without storage) is set so that it produces no electricity at night. The model then requires the system to invest in other technologies to produce in the evening etc. This increases the NPV to a level wherein solar is not invested. PIDA- Africa energy outlook 2040 consists of an inventory of power plant projects in the different African power pool consortia. The outputs from the modelling exercise behind the PIDA outlook are not available as open data (Sofreco consortium, 2012). PIDA+ is an inventory created for this study, which includes PIDA information (whatever was available from the reports and discussions), data from the latest versions of the regional master plans and stakeholder engagement meetings. Hence, to clarify the reviewer's comments, PIDA+ is an inventory of an expert elicitation process and our optimization results cannot be directly compared with the PIDA+ inventory.
2.2 The treatment of weather-dependent variability of renewable generation and demand in the capacity extension model is not explained sufficiently. Past work (e.g. Bloomfield et al., doi: 10.1088/1748-9326/11/12/124025) has shown how important this variability is for the results of power system models irrespective of future climate variability. How does the time slice approach of 12 months and 2 day-types consider weather-driven inter-annual variability of electricity demand and of supply from variable renewable generation technologies like PV and wind?	2.2 This is an important comment. This study does not focus on several climatic and subsequent weather impacts on the energy system. Examples of those include: wind, solar (as noted by the reviewer), as well as HVAC, water pumping and others. We highlight this limitation. The study, however, does focus on changes in hydro generation. Thus simplified representation of other parts of the system are made to derive limited tractable insight. The seasonal weather patterns for precipitation are calculated at monthly time-step, translated into water flows and those are used to determine potential hydro generation estimates for their respective climates. To summarise those simplified representations: The Eastern African Power pool (EAPP) model has 48 splits in each modelled year; each year is divided into 12 months and each month is further classified into 4 parts. The splitting of the time steps is explained as part of the supplementary material

	(section 1.2.2). Since the focus of this paper is the climate induced hydropower variability and its impact on the overall electricity sector, we have not considered changes in intra annual variability for solar and wind as a function of climatic change. Average daily load curves for solar PV output (both PV and PV + storage) and total annual maximum renewable generation potentials, used in this study, are based on IRENA (2014). Though we have not focused on the weather dependent nature of generation and demand – we agree that it is important. (A note pointing to this is added – with reference to the uncertainty that it introduces).
2.3 In Figure 4, the no-adaptation electricity generation costs for Tanzania and Uganda fluctuate quite a bit through time. This suggests to me that there in addition to the climate signal there is substantial year-by-year variability in the data that the authors do not address, but which could have a strong influence on results.	2.3 In our opinion, in this model, the fluctuations are solely due to climate variability. When there is a reduction in hydropower generation, the only available options are expensive but quickly deployable, fossil fuel based generation options; hence the fluctuation in the annualised cost of electricity generation. The only other year-by-year variability in the data is the fossil fuel prices (from power pool outlooks).
2.4 It is also not clear to me why costs should increase through time. Given the falling costs of major technologies, it would seem more sensible that generation costs fall with time.	2.4 In this study, for all the generic power plants, we consider a capital cost that does not change over time in the modelling system. The values correspond to the technology-specific starting value, in all the four different scenarios mentioned in the IRENA report (which is a source for generic capital costs). Concerning the increasing costs, it corresponds to the usage of fossil fuels in the respective countries, and the fossil fuel prices increase over time. The countries use the cheapest options available to them (bulk hydro) and then move to the more expensive options when those are depleted.
3. Scenario results and selection of scenarios to compare in the paper 3.1 As far as I understand, the EAPP electricity sector planning model is run for the 6 selected climate realizations (perfect foresight model runs), then the model is run with each of the 6	3.1 The reviewer’s understanding is correct with respect how the scenarios are formulated. A slight correction with respect to the number of scenarios. There are 7 climates in total (6 projections and one derived from historical values); So there are 7 PF runs. Each of the 7 optimal strategies (corresponding to each climate) are run using climate information from 6 other climates. Hence, 49 scenarios in total are analysed. In line with the

resulting optimal expansion plans but using climate data from the 5 other scenarios. In addition the no-adaptation expansion plan (optimized for current climate) is also compared to the 6 future climate scenarios, resulting in a total of 42 combinations of electricity sector expansion plans and climate conditions. From these, 2 sets are chosen for each country: the most resilient adaptation strategy of the 6 planning strategies considered, and the no-adaptation expansion operated under the driest climate conditions of the 6 future climate scenarios. Is this interpretation correct? I think it would be nice to have a table somewhere explaining these different scenarios.	reviewer's comments, out of the 49, the most resilient and the no-adaptation strategy for the driest climate are presented as results in Figure 5. In addition to the description of types of optimization runs in the methods section, we have added a paragraph (section 2.1) in the supplementary material to address the reviewer's comments. Tables 2 and 3 of the supplementary material give an overview of the scenarios that were analysed.
3.2 Little concrete detail is given on the two criteria used to select the most resilient adaptation strategy. These should be described in more detail. See also the comment above on how this study relates to the World Bank report cited as ref number 4	3.2 We have added some extra information on the criteria used in this study. Sections 1.5.2 and 2.3 in the supplementary material have been improved to reflect on the evaluation criteria.
3.3 If the authors do not intend to release the full results, they could show figures or tables of results of the perfect foresight capacity expansion model runs for one or two countries to show the range of capacities that result in different electricity generation technologies.	3.3 Many thanks for the useful suggestion. The supplementary material (section 2.2.1), now, contains country-specific expansion plans (generation and capacity) for all the PF scenarios. Please note that the full model code is also provided on GitHub, should analysts wish to use it.
4.Uncertainty of other factors vs uncertainty in climate 4.1 One of the key results seems to be on p.7: "Uganda and	4.1 The variation in costs are reported for the cumulative period between 2017-2050. We have restructured the sentence in the main document to make the point clearer. Thank you for noting the lack of clarity.

Tanzania could expect costs to vary between -18% and +28%, -5% and +23% respectively between the wettest and driest climates when compared to the baseline" (the authors should clarify though whether this statistic is for 2050 or across some average of years). While these are substantial numbers, they do beg the question what the variability introduced by other factors would look like.	
4.2 The authors state in the SI, last para of section 1.2.1, that because of the considerable uncertainty in projecting energy demand "the results need to be analysed with a grain of salt." However, I do not see this uncertainty really being dealt with in the study. There seems to be a single demand projection for each country in the EAPP. For example, electricity demand in Tanzania is projected to grow from 7.5 TWh in 2015 to 69.73 TWh in 2050 according to Table 5 in the SI appendix. How would the results of this study change if demand were only half as high, or double that value, in 2050? The possible uncertainty seems enormous and because the paper does not present much detail on the different sets of electricity generation technologies deployed under the different climate scenarios, the reader cannot judge whether the uncertainty introduced by demand (i.e., things like economic and technology development) would dwarf the uncertainty introduced by climate through 2050. In particular, given that in the scenarios presented here the vast majority of electricity demand across EAPP is satisfied by gas and coal generation in 2050, coal and especially gas prices would seem like such a major source of price volatility that they might also dwarf the effect of different climates. How, if at all, does the study consider this? The second para on p.10 states: "These results emphasise the message that irrespective of the climate, availability of	4.2 Again, this is an important point. However, we would respectfully note that our analysis is not designed to provide a comparative assessment of uncertainty. We aim to make a limited contribution. The objective of this study is to assess the resilience of the power sector to the effect of a changing climate on hydro generation. Thus we deliberately chose to keep much constant in this piece to help with the tractability of insights. There are a number of important uncertainties can be included in the future analysis – they may as the reviewer points out dwarf climate impacts. Others of particular interest include fuel costs, efficiency, demand, trade, emissions constraints and technology development. A comparative assessment of uncertainties is made to aid policy making under uncertainties. The study makes it's limited (but first of a kind for the region) contribution in describing dynamics around climate uncertainties induced by weather patterns affecting water flows and subsequently hydro generation.

electricity trading infrastructure stands to play a significant role in balancing electricity prices within the EAPP." The authors could expand on this to provide a better overview of how large an effect climate uncertainty has on the other major sources of uncertainty in the development of the EAPP economies and their power systems.	
Specific comments	
The paper would benefit from a thorough proofreading to improve the quality of English.	We have proofread the paper and have modified the text to improve the language.
Introductory paragraph: It seems strange to have only a single reference, and that reference to be to the Platts database. I would expect that to be to a definition of EAPP.	We have added a reference for the EAPP in the introductory paragraph. The platts database was referenced to justify the heavy gas based electricity generation capacity in the region.
Footnotes are not used in Nature journals as far as I know, but in any case, they should not use the same numbering and style as endnotes - the two are currently indistinguishable in the text. For example, on p.5 you write "... the PIDA+4,23 expansion plan" where 4 seems to refer to a footnote and 23 to a endnote. You should explain what PIDA+ is in directly in the main text.	We have removed the footnotes from the main and supplementary document and incorporated the information as text, wherever appropriate.
Figure 1 label is unclear. I suggest something like: "Projections of mean annual CMI across Africa's major river basins under different radiative forcing scenarios. The CMI (Climate Moisture Index) is a measure of aridity [...]"	We have corrected the caption as per the reviewer's suggestion.
Para 2 on p.5 states that the EAPP power system expansion model contains "all individual hydropower plants, which were represented in the water model". A map of these plants would be nice to see, but more importantly, some summary statistics on the variability of electricity generation of these different plants under the different modelled climate future scenarios. That would help the reader better understand the value of considering each individual hydro plant in this study, i.e. whether the modelling methods used are actually able to actually represent local variability in hydro generation.	A new section has been added in the supplementary material to emphasise the need for site-specific hydropower representation. The reasons behind the site-specific approach are:  • Site-specific costs differ according to location, type and size of the plants • Site-specific, competing uses of water resources affect hydropower generation • Climate variability: hydropower plants in the same region/country could be part of different sub,-basins of the Nile River; hence, a

	variation in the climate and land use induced runoff could be different. New figures showing the variation in monthly electricity generation from different hydropower plants across all the analysed climates has been added to section 1.4 of the supplementary material (for years, 2020, 2030, 2040 and 2050).
Figure 3 needs a better color scheme. It is almost impossible to tell the colors for diesel and geothermal apart. Perhaps 5-year increments rather than plotting each individual year would make the figure less busy and easier to read? Also, the figure caption is not clear. Do all four panels show the cost-optimal expansion plans (for either EAPP or the three individual countries) under the baseline (historical) climate? Why were these three countries selected?	The caption and colour scheme for figure 3 have been modified. We agree that 5-year increments would make the figure easier to read, but hydropower variability between years, which affects the cost of electricity generation is eclipsed by stepping every 5 years. Hence, we would prefer to maintain the yearly representation. However, we have modified the figure dimensions to attempt to derive an easier to absorb representation. Yes, the panel shows cost optimal expansion plans for EAPP and three other countries for the baseline (historical) climate. We chose Uganda, Tanzania and Ethiopia because they have a high share of hydropower in their electricity systems and a significant share of this hydropower is expected to be installed in the next 10-15 years. Also, we have added generation mix and capacity results for all the countries and climates in the supplementary material.
Can you add the country-specific results analogous to Figure 3 for the remaining countries as supplementary figures? It would be interesting to see what the expansion scenarios look like for deployment optimisation runs other than the baseline.	Yes, all the results are now available in the updated supplementary material.
Figure 4 needs a better caption. You should describe all three panels in the caption so that the figure can be understood without searching for where it is described in the text. Perhaps here you could also include the deployed generation technology mix (maybe in 5 year increments to keep the figure readable) for comparison?	We have modified Figure 4's caption and the corresponding text in the main paper. We have included a graphic in the supplementary material, where hydropower generation (in selected years) is plotted along with the annualised cost of electricity generation. In this figure, we have illustrated only the hydropower generation as the shares of it varies across countries and plotting the entire generation mix (across different scenarios) made the graphs slightly difficult to read. However, as mentioned in the previous comment, we have added country-specific generation mix in the

	supplementary material for all the countries and all the climates to address the reviewers' concerns.
Figure 5 needs a more detailed label to explain the difference between worst case and most resilient strategy.	We have expanded the caption to make the message clearer
Figure 6: Visually interesting, but I do not see any explanation of the bar chart alongside the outer circumference of the circles. What are these percentage values showing?	We have modified the caption to explain the circumferential bar charts.
Comments on the Methods section	
First para of methods on p.11 has two consecutive sentences starting with "The EAPP model " - I think the second of these should be "The WEAP model" (maybe a bad idea to give these models such similar names? It certainly makes reading the manuscript an exercise in concentration to make sure one does not mix up these two).	We thank the reviewer for the suggestion. We have changed the naming of the EAPP model to the energy model and the WEAP model to water model to reduce ambiguity and confusion while reading the article.
You state that over 500 electricity generation technologies were included, can you clarify what these >500 technologies are?	We thank the reviewer for pointing out this misrepresentation. In OSeMOSYS, the framework used to develop the electricity sector expansion model, entities that generate and transfer commodities like electricity and fossil fuels are called technologies. In this model for the EAPP, each country had more than 50 technologies consisting of different electricity infrastructures (as mentioned in table 12 of the supplementary material). These include the following  • Site-specific large hydropower plants and grouped micro/small hydro plants • Other non-hydro electricity generation plants (Geothermal, Solar PV: rooftop, grid-connected, rooftop with Battery, Wind, Natural gas turbines (CCGT/OCGT), Nuclear, Diesel plants (distributed and grid connected), biomass, Coal plants, Heavy fuel oil (HFO) based plants • Transmission and distribution (T&D) infrastructure • Fossil fuel import and local extraction options

	Hence, in total, the technologies in the model amounted to more than 500. The complexity in any such modelling framework (not only OSeMOSYS) increases with the number of represented technologies and hence the significance. We have updated the main document to reflect the above discussed representation.
Is technological learning considered at all in the model? It would seem very likely that the cost of PV and in particular of batteries continue falling substantially and that by 2050 the relative cost difference between PV and e.g. coal will look very different than what table 12 now states (see e.g. Schmidt et al, http://dx.doi.org/10.1038/nenergy.2017.110).	We agree with the reviewer’s opinion that the capital cost of Solar PV is expected to decrease. And it is likely to be competitive with hydropower-based generation. Though battery prices may increase due to competition for resources, and integration costs for solar may remain a barrier. However, according to the renewable promotion scenario in the aforementioned IRENA study (IRENA, 2013) where a technology learning curve is applied, and the capital cost of non-hydro renewables decrease over time), even by 2030, the cost (levelised cost of electricity generation (LCOE)) of grid-connected PV is expected to be 39% more than that of hydro-based electricity generation. The share for rooftop PV is expected to be 56% more in 2030. We agree that this share is likely to decrease and by 2040, it could compete with hydropower and this is something to be taken into consideration while drawing conclusions from the results of the modelling period.
It seems unrealistic to assume that only currently planned transmission capacity will exist in 2050. Can this assumption be justified? How different would model results look if the planning model were allowed to increase the transmission capacities between countries?	This study considers existing, planned (under construction) and identified future un-committed transmission infrastructure as mentioned in the master plan of the EAPP. Our analysis simply asks ‘What is the study outcome?’ holding this assumption. That said, most of the countries in the EAPP plan to be able to meet the bulk of their national demands in the case of transmission interruptions. We do not attempt to predict the future, but unconstrained development may lead to even more unrealistic outcomes. In the European Union (EU), where the policies aim to connect all the constituent countries into one single market with reserves of different technology types in multiple countries, it is a long shot and expected to be achievable only around 2050, with an expected investment of 300-450 billion Euro (European

	Commission, 2017). Hence, we would argue a fully interconnected future in the EAPP is unlikely before 2050. A recent study, by some of the authors of this article, (Taliotis et al., 2016) explores the impact of enhanced trade scenario for the entire African continent. The inter-country trade capacity in the EAPP countries reported by Taliotis et al. for 2040 is twice that of the value from our study. Moreover, in our estimates, by 2040, the trade infrastructure capacity increases four folds compared to the existing installed capacity.
Can you clarify the reason for also including older CMIP3 data as inputs to the WEAP model? To me it would make a lot more sense to limit yourself to CMIP5, instead consider more than two RCPs, and possibly show data split by RCP rather than between CMIP3 and CMIP5 (in Figure 1).	The authors agree that the CMIP5 runs are more relevant given their more recent release. However, this study relies on both CMIP3 and CMIP5 runs to expand the range of potential climate change impacts and adaptation options evaluated. As a result, this figure presents CMIP3 runs to illustrate differences between the ensembles used in AR4 and AR5.
The authors only consider electricity demand and supply. How do other energy-using sectors factor in to the analysis, for example mobility?	The analysis of the transport sector lies outside the scope of this study, as the objective of this analysis was to analyze the vulnerability of the power sector to a changing climate. That said, the point is essential. It raises again the reviewer's sage observation relating to the need for a comparative assessment of uncertainties and their impact.
Supplementary material: Referencing the SI would be easier if all items were numbered consecutively with Sxxx, for example, Table S11 rather than Annex A1-Table 11. Also the SI should have page numbers.	We have modified the numbering as per the reviewer's suggestion
Appendix Table 11 has no source.	The source has been added
Tables 6 and 7 in the supplementary appendix are unclear. What are the four rows labelled Part 1 - Part 4?	The four rows labelled part 1-4 are the four splits in each month. Table 6 gives the split in the actual time. Table 7 is the share of the total demand that needs to be satisfied in the respective time slices. We have changed the description of the tables to clarify the row names.

Reviewers' comments:

Reviewer #1 (Remarks to the Author):

I am pleased with the answers and the additional information provided by the authors. I therefore recommend that this manuscript be published as it is.

Reviewer #2 (Remarks to the Author):

The authors replied to the comments and made most of the required revisions.

I'm only concerned by the fact the authors consider a constant cost for the technologies. Their arguments are not entirely convincing. In some European countries, the renewable energies drop of cost and their deployment have jeopardized the future of hydropower. Developing a robust decision-making model while discarding a significant driver and a major source of uncertainty remain questionable. Which conclusions remain relevant if we consider this driver and its related uncertainty?

As the article mentions the issue, I think the paper can be published but would benefit from a substantial discussion. The authors carried out interesting research, which contributes to a topical issue.

Reviewer #3 (Remarks to the Author):

I thank the authors for their edits in response to my comments. Most of my points are addressed. What remains open is outlined below.

My main remaining issue is with respect to my comment 2.3: The authors state that the only source of year-by-year variability is the climate signal, but acknowledge that the energy model deals with variability due to climate primarily with the use fossil fired generation, and that fuel costs also vary year on year.

First, for this explanation to be convincing, I am still missing the fuel price data and how it is projected to change in the future. Does fuel price linearly increase to some future projected value, or does it fluctuate? The authors say that fuel prices are in the supplementary material but I cannot find the information there. There is a Table S9 (cost of domestic fuel extraction), but it is not referred anywhere, so it is not clear how these data are used in the study, and it is not stated which year these costs are from (in any case, they seem to be static, not time dependent). As the model+data used to generate the results reported in this study are not available, there is no way to understand how or even whether the data listed in the supplementary information are implemented in the model.

Second, if the aim of the study is to understand the climate signal, should fuel costs not be held constant? In particular since the authors also clarify in response to my comment on technological learning that they do not consider any other cost changes (such as technological learning). If all other costs relating to electricity generation are constant through time, why do fuel costs change?

Some other minor points remain:

With respect to my comment 3.1, for consistency, tables 2 and 3 in the supplementary should also be labelled S2 and S3.

On p.14, I still do not understand why you say the model considers 500 technologies. Table S12 referred there lists 25 technologies (by the way, the fact that the table caption for Table S12 is to the right of the table rather than below, is a bit confusing -- this is also the case with some other tables in the supplementary information). How do these 25 and the 500 relate? Also, you now write "48-time steps" in that section which I assume is a typo.

Figure 3: You have replaced "EAPP" with "Egypt" in the figure but did not update the caption accordingly.

Figure 5: Labels overlap the bars. Possibly they should be repositioned.

On reading some of the responses to my comments, clarifying that the sole aim of the paper is to assess the resilience of the electricity sector to the effect of climate change on hydropower, I think that title could be improved. I.e. something like "Resilience of the Eastern African electricity sector

to climate-driven changes in hydro power generation". Given that the resilience of no other electricity supply infrastructure component is assessed - e.g.: thermal power plants, the grid, solar power, wind power, that would seem to more clearly reflect the paper's content than the current title, "Resilience of Eastern African Electricity Supply Infrastructure to Climate Change".

I acknowledge the fact that part of the code used is openly available, and thank the authors for linking to the GitHub repository for the OSeMOSYS code. However, my point was that the specific code and data used for this study, as well as the setup with linked models, is not available for review (nor for others to later expand on this study). This is just to comment on that fact. I request no action from the authors on this point. The journal does not require code or data availability, so there is no reason for the authors to make any of these things publicly available.

Resilience of Eastern African Electricity Supply Infrastructure to Climate Change

Response to Reviewers

The authors wish to thank the reviewers for their second round of feedback that has helped to improve the quality of the paper. The comments from the reviewers and the corresponding response and actions taken are provided below.

Reviewer #1 (Remarks to the Author):

I am pleased with the answers and the additional information provided by the authors. I therefore recommend that this manuscript be published as it is.

Author response

We are happy that we were able to address your comments and improve the quality of the manuscript.

Reviewer #2 (Remarks to the Author):

Comment from Reviewer #2	Response-Action
The authors replied to the comments and made most of the required revisions. I'm only concerned by the fact the authors consider a constant cost for the technologies. Their arguments are not entirely convincing. In some European countries, the renewable energies	In response to the reviewer's comments, we have included a discussion piece in the supplementary material (section 3) along with a summary of it in the main article. To support this discussion, we made a new model run, using OSeMOSYS, of the Eastern African Power Pool (EAPP). This new run included cost reductions due to technology learning for renewable technologies. Africa specific projections from the New Policy scenario of the World energy outlook 2016 (WEO 2016) were used for this exercise. All the other costs and assumptions remain the same as in our original reference scenario.

Comment from Reviewer #2	Response-Action
drop of cost and their deployment have jeopardized the future of hydropower. Developing a robust decision-making model while discarding a significant driver and a major source of uncertainty remain questionable. Which conclusions remain relevant if we consider this driver and its related uncertainty? As the article mentions the issue, I think the paper can be published but would benefit from a substantial discussion. The authors carried out interesting research, which contributes to a topical issue.	Indeed, the introduction of learning rates in renewable technologies does bring some change in the electricity generation capacity of the power pool. Natural gas and—to some extent—coal-based electricity generation is replaced by a mix of centralised and distributed solar PV technologies. By 2050, approximately 85 GW of gas and 36 GW of coal infrastructure is replaced by ~ 180 GW of grid-connected and decentralised solar PV. It is interesting to note that, the output from hydropower is practically not affected, as it is still a more economic option than solar power, in the long term. This is of significance as the bulk of trade in the region is propelled by inexpensive hydropower in water resource-rich countries. Hence, to reflect on the reviewer’s comment on the relevance of the conclusions under this change:  1. Yes, the cost of electricity generation will be affected by the change in investment costs for renewables. The reduction in total system cost due to technology learning is expected to be about 150 billion USD₂₀₁₀ for the period between (2017-2050). But, we must also understand the instability that it brings to the grid and the increasing need for storage technologies in the system. Without reserve power plants and sufficient storage in the grid, this expansion of solar power might not be realistic. 2. The learning rates are expected to bring some change in the trade dynamics. However, trade based on inexpensive electricity generation from hydropower continues to play a significant role in the power pools generation mix. Of course, the effects of a progressively dry climate could result in higher—non-hydro—renewable penetration, which needs to be explored in detail

Reviewer #3 (Remarks to the Author):

I thank the authors for their edits in response to my comments. Most of my points are addressed. What remains open is outlined below.

Comment from Reviewer #3	Response-Action
My main remaining issue is with respect to my comment 2.3: The authors state that the only source of year-by-year variability is the climate signal, but acknowledge that the	The authors thank the reviewer for bringing to our attention the issue with the fossil fuel prices. They were obtained from the Eastern African power pool’s (EAPP) master plan (EAPP, 2011). We have updated table S9 with the yearly

Comment from Reviewer #3	Response-Action
energy model deals with variability due to climate primarily with the use fossil fired generation, and that fuel costs also vary year on year. First, for this explanation to be convincing, I am still missing the fuel price data and how it is projected to change in the future. Does fuel price linearly increase to some future projected value, or does it fluctuate? The authors say that fuel prices are in the supplementary material but I cannot find the information there. There is a Table S9 (cost of domestic fuel extraction), but it is not referred anywhere, so it is not clear how these data are used in the study, and it is not stated which year these costs are from (in any case, they seem to be static, not time dependent). As the model+data used to generate the results reported in this study are not available, there is no way to understand how or even whether the data listed in the supplementary information are implemented in the model.	values. The projections from the EAPP master plan extend until the year 2038. From then, until 2050, we assume it to remain constant. The prices from the master plan show a linear trend for diesel oil (IDO), heavy fuel oil (HFO) and natural gas but have a stepwise increase for coal.
Second, if the study aims to understand the climate signal, should fuel costs not be held constant? In particular since the authors also clarify in response to my comment on technological learning that they do not consider any other cost changes (such as technological learning). If all other costs relating to electricity generation are constant through time, why do fuel costs change?	In this study, since the focus is on the EAPP, we decided to adopt the fossil fuel price projections from the power pool's official master plan. Please also see the relevant action taken in response to a comment by reviewer #2. A new model run with technology learning rates has shown that hydropower investments are unaffected (for the reference scenario), as the increased rate of renewable energy adoption substitutes fossil-fired generation. Hence, the insights offered regarding climate impact on hydropower output remain valid in both instances (with or without learning rates). While this analysis could - and we recommend should - be extended to include a larger range of uncertainty, we urge caution. The structure of uncertainty is different; we suspect that simply running a large ensemble of scenarios might result in a loss of important insight. Consider technology costs and fossil fuel price projections.

Comment from Reviewer #3	Response-Action
	On the one hand, renewable technology costs are often driven by technological improvements, competitive procurement opportunities, and the presence of a large base of experienced project developers, to name a few (IRENA, 2018). On the other hand, fossil fuel costs are driven by factors like the availability of storage/inventories, local and international geopolitical situations, the cost of resource recovery, transportation costs and weather, to name a few (Atalla et al., 2017; Holditch and Chianelli, 2008). Fossil fuel price projections are often carried out using market models considering global fossil fuel reserves and the use of such a model was outside the scope of this study. The shape of each uncertainty—and the futures in which they exist—might be inconsistent. This would require the development of methods to consistently generate them, with screening out inconsistencies a priori or ex-ante. For both, we suspect non-trivial consideration is needed.
Some other minor points remain: With respect to my comment 3.1, for consistency, tables 2 and 3 in the supplementary should also be labelled S2 and S3	We have updated the table caption names to be consistent with the rest of the document.
On p.14, I still do not understand why you say the model considers 500 technologies. Table S12 referred there lists 25 technologies (by the way, the fact that the table caption for Table S12 is to the right of the table rather than below, is a bit confusing -- this is also the case with some other tables in the supplementary information). How do these 25 and the 500 relate? Also, you now write "48-time steps" in that section which I assume is a typo.	In most of the energy systems optimisation models (including OSeMOSYS), each entity that generates or transmits energy is called a technology. In the EAPP model, each electricity generation technology in each country is regarded as one technology that can contribute to the generation mix. For instance, the coal power plants in Kenya and Tanzania, even though identical technology per se, they are represented as two separate technologies in the model. Likewise, each country has the option of generating electricity from a suite of technologies, taking into consideration: costs, efficiencies and other resource constraints. Also, since we had represented many large hydropower plants as individual technologies, the total number of technologies in the model accumulated to about 500. This count includes energy transmission and distribution infrastructure. The complexity of these models increase with a larger technological representation, and hence the mention.

Comment from Reviewer #3	Response-Action
	Regarding the temporal resolution, each year in this model is divided into 12 months. And each month is further broken down into four parts. Hence, each year is effectively split into 48 (12*4) time slices (time steps). Supplementary section 1.2.2 explains the temporal resolution of the model. The size of the matrix to be optimised increases exponentially with the increase in the number of time slices. Hence, the significance in mentioning them. We have corrected the table captions in the document. We thank the reviewer for bringing them to our attention.
Figure 3: You have replaced "EAPP" with "Egypt" in the figure but did not update the caption accordingly.	We have corrected the figure title to match the caption.
Figure 5: Labels overlap the bars. Possibly they should be repositioned.	We have modified figure 5 to remove the overlaps.
On reading some of the responses to my comments, clarifying that the sole aim of the paper is to assess the resilience of the electricity sector to the effect of climate change on hydropower, I think that title could be improved. I.e. something like "Resilience of the Eastern African electricity sector to climate-driven changes in hydro power generation". Given that the resilience of no other electricity supply infrastructure component is assessed - e.g., thermal power plants, the grid, solar power, wind power, that would seem to more clearly reflect the paper's content than the current title, "Resilience of Eastern African Electricity Supply Infrastructure to Climate Change".	As per the reviewer's suggestion, we have updated the title to "Resilience of the Eastern African electricity sector to climate-driven changes in hydropower generation."
I acknowledge the fact that part of the code used is openly available, and thank the authors for linking to the GitHub repository for the OSeMOSYS code. However, my point was that the specific code and data used for this study, as well as the setup with linked models, is not available for review (nor for others to later expand on this study). This is just to comment on that fact. I request no action from the authors	Since, this article focusses only on the power sector, the data file used for the EAPP OSeMOSYS model has been made available in a Zenodo data repository (10.5281/zenodo.1478149). All the other relevant data that went into the model development is available in the annexe to the supplementary material.

Comment from Reviewer #3	Response-Action
on this point. The journal does not require code or data availability, so there is no reason for the authors to make any of these things publicly available.	

Bibliography

- Atalla, T., Blazquez, J., Hunt, L.C., Manzano, B., 2017. Prices versus policy: An analysis of the drivers of the primary fossil fuel mix. *Energy Policy* 106, 536–546. <https://doi.org/10.1016/j.enpol.2017.03.060>
- Holditch, S.A., Chianelli, R.R., 2008. Factors That Will Influence Oil and Gas Supply and Demand in the 21st Century. *MRS Bull.* 33, 317–323. <https://doi.org/10.1557/mrs2008.65>
- IRENA, 2018. Renewable Power Generation Costs in 2017 (No. ISBN:978-92-9260-040-2). International Renewable Energy Agency, Abu Dhabi.
- EAPP, 2011. EAPP Master Plan (Grid code study). SNC LAVALIN INTERNATIONAL INC, PARSONS BRINCKERHOFF.

REVIEWERS' COMMENTS:

Reviewer #2 (Remarks to the Author):

From my perspective, the paper can be published.

Reviewer #3 (Remarks to the Author):

The authors have addressed all points and I have no further comments